# Heterogeneous Transfer Learning with Feature Transformation-Based Adaptation for Modeling Dynamical Systems

## Abstract

In this work, a novel heterogeneous transfer learning framework is proposed for modeling dynamical systems, where the source and target domains have different feature spaces. A feature transformation scheme is implemented via customized adaptation layers integrated into the pre-trained model. We conduct theoretical analysis of heterogeneous domain adaptation, demonstrating the generalization performance of the pre-trained model on the target domain after feature transformation. Based on this analysis, a two-phase training strategy is proposed to improve the performance of the heterogeneous transfer learning model. The experimental results in four case studies across different application domains demonstrate the effectiveness of the proposed method.

## 1 Introduction

Accurate modeling of dynamical systems remains a long-standing challenge across many engineering domains. Reliable models not only yield valuable insights into complex dynamic behaviors, but also serve as a critical foundation for downstream tasks such as system operation and control. However, capturing the behavior of real-world systems is particularly difficult due to their intricate nonlinearities and tightly coupled subsystem interactions. A variety of data-driven methods, such as machine learning techniques, have been proposed to address this problem. However, collecting sufficient data in practice is often time-consuming and costly, especially for complex nonlinear systems, where traditional machine learning methods typically require large datasets to achieve high modeling accuracy. Transfer learning is a powerful tool to address the data scarcity problem by taking advantage of knowledge in the source domain to improve performance in the target domain (Neyshabur et al., 2020; Zhuang et al., 2020). Despite its successful applications (Theodoris et al., 2023; Jiang et al., 2022; Zhao et al., 2024; Li et al., 2022), the performance of conventional transfer learning methods is based on the key assumption that a source process is available with a similar configuration (e.g., same feature space) and sufficient data. This assumption restricts its applicability, particularly in modeling nonlinear dynamical systems, where identifying such a source process is often challenging in practice.

Heterogeneous transfer learning (HTL) aims to improve performance in the target domain by utilizing knowledge from the source domain with a different feature space. By accommodating differences in feature representations, HTL enables broader applicability across diverse tasks where source and target domains are inherently mismatched. The feature mismatch between the source and target domains can be addressed using the feature transformation method. Recent years have witnessed a flourishing in research on HTL methods in various areas (Day & Khoshgoftaar, 2017; Bao et al., 2023). However, designing effective HTL frameworks for modeling dynamical systems is inherently challenging. Specifically, state variables commonly used as features are often strongly coupled in nonlinear systems, leading to significant discrepancies between the source and target domains with different feature representations. Furthermore, efficiently adapting the pre-trained model to the target domain with limited data remains a challenge, especially when the pre-trained model is a neural network. Therefore, it is important to develop an HTL framework that incorporates proper feature transformation and efficient adaptation to model nonlinear systems with limited target data.

Theoretical analysis of the generalization error in domain adaptation can serve as a powerful tool to guide the design of transfer learning methods. Specifically, the generalization performance of the pre-trained model from the source domain on the target domain indicates whether the pre-trained model can provide a good initial guess for the target task. Several results on the generalization error bound for domain adaptation have been published (Zhang et al., 2019; Redko et al., 2020). For example, in Mansour et al. (2009) the generalization bound for domain adaptation was investigated using Rademacher complexity theory. However, deriving the generalization error bound for heterogeneous domain adaptation is particularly challenging due to mismatched feature spaces. Theoretical analysis of heterogeneous domain adaptation must account not only for the discrepancy between source and target domains but also for the effect of feature transformation methods designed to address the feature mismatch.

Motivated by the above considerations, we develop a heterogeneous transfer learning framework to address the modeling problem for dynamical systems. The generalization performance of heterogeneous domain adaptation is theoretically analyzed using statistical learning theory. Based on this theoretical understanding, the training procedure of the pre-trained model is designed. Specifically, the feature transformation in this work is implemented via the designed adaptation layers. The major contributions of this work can be summarized as follows:

- We derive the generalization error bound for heterogeneous domain adaptation with feature transformation, providing a theoretical guarantee for modeling dynamical systems via HTL methods with neural networks.

- We propose a novel two-phase HTL algorithm based on the theoretical analysis. Specifically, transformation matrices are learned via customized adaptation layers to address feature mismatch.

- Experiments on several case studies, including chemical engineering, process systems engineering, and robotics, demonstrate the effectiveness of the proposed framework.

## 2 RELATED WORK

### 2.1 HETEROGENEOUS TRANSFER LEARNING

Unlike homogeneous transfer learning, HTL methods focus on cases where the source and target domains have different feature spaces. These methods typically employ data- or model-based techniques to address the feature mismatch. Data-based methods transfer source data into the target domain (Wu et al., 2019; Zhu et al., 2011; Li et al., 2019). For example, in Zhu et al. (2011), a matrix factorization method was proposed to generate latent features for effective knowledge transfer. Li et al. (2019) introduced a deep matrix completion framework to address heterogeneous features across domains. Model-based methods aim to adjust model structures or parameters. In particular, adaptation layers have been incorporated into pre-trained models—especially large language models—to address feature heterogeneity (Guo et al., 2023; Qi et al., 2020). HTL methods have been successfully applied in various domains, including natural language processing (Zhou et al., 2016), image classification (Zhu et al., 2011), protein prediction (Han et al., 2025), and manufacturing (Kevin et al., 2021).

Currently, several HTL methods have been proposed for time-series data. While they achieve good performance, most focus on classification tasks (Fang et al., 2022; Liu et al., 2023) and rely on assumptions that are difficult to satisfy in nonlinear dynamical system modeling, such as instance correspondence (He et al., 2020) or multiple sources (Li et al., 2019; Wilson et al., 2023). In parallel, various approaches have been developed to improve generalization and robustness in dynamical system modeling (Yin et al., 2021; Kirchmeyer et al., 2022; Qin et al., 2025; Mouli et al., 2024). However, these works primarily address heterogeneity arising from mismatched parameters, often assuming that the dynamics can be decomposed into a shared fixed term and an environment-specific term (Yin et al., 2021; Kirchmeyer et al., 2022). Such assumptions may not hold in the heterogeneous feature setting considered here, particularly when the source and target domains involve different state variables.

## 2.2 THEORETICAL ANALYSIS ON DOMAIN ADAPTATION

Since knowledge from the source domain is utilized to develop a model for the target domain, analyzing the generalization error of a pre-trained model on the target domain is a critical tool for improving the transfer learning performance with theoretical guarantees. Early theoretical results on domain adaptation can be found in Ben-David et al. (2006); Mansour et al. (2009); Cortes & Mohri (2011), and a comprehensive overview is presented in Redko et al. (2020). The key challenge in the theoretical analysis of HTL lies in the mismatched features, which is not fully addressed yet. Zhou et al. (2019) proposed a sparse heterogeneous feature representation method, and provided its generalization error bound in a multi-class setting. In Fang et al. (2022), two semi-supervised HTL methods were proposed based on the kernel and neural network models with theoretical guarantees. Chang et al. (2024) provided the statistical estimation guarantees for heterogeneous domain adaptation in regression tasks with feature mismatch.

## 3 PROBLEM FORMULATION

### 3.1 CLASS OF NONLINEAR SYSTEMS

In this work, we consider a class of nonlinear systems, of which the dynamics can be represented via the following ordinary differential equation (ODE):

$$\dot{\mathbf{x}} = f(\mathbf{x}, \mathbf{u}) \tag{1}$$

where $\mathbf{x} \in \mathcal{R}^{d_{xv}}$ and $\mathbf{u} \in \mathcal{R}^{d_u}$ denote the state and input vectors, respectively. $\dot{\mathbf{x}}$ is the time derivative of the state vector.

### 3.2 PROBLEM STATEMENT

The objective of this work is to model the nonlinear system of Eq. 1 with limited data under the heterogeneous transfer learning method. While the model of Eq. 1 is expressed in continuous-time form, practical modeling typically involves constructing a discrete-time model that predicts the state at the $t$th time step (i.e., the model output) based on the current state measurements and inputs up to time step $t$ (i.e., model input). We consider the case where both the source and target processes can be described by Eq. 1 but are characterized by different state variables (e.g., $\mathbf{x}$ could represent different physical variables). We use $\mathcal{S}$ and $\mathcal{T}$ to denote the source and target domain, where the source domain has sufficient data and the target domain has only limited samples. As a result, the feature space of $\mathcal{S}$ is different from that of $\mathcal{T}$, motivating the development of an HTL-based modeling approach to enable knowledge transfer across domains.

## 4 FEATURE TRANSFORMATION

The source and target domain can be described as follows:

$$\mathcal{S} \in (X_S \times Y_S), \quad \mathcal{T} \in (X_T \times Y_T) \tag{2}$$

where $X_S \in \mathcal{R}^{d_{xs}}$, $Y_S \in \mathcal{R}^{d_{ys}}$, $X_T \in \mathcal{R}^{d_{xt}}$, and $Y_T \in \mathcal{R}^{d_{yt}}$. $X_S$, $Y_S$ and $X_T$, $Y_T$ denote the model input and output features of the source and target domains, respectively. Such features can be constructed from the input, state, and output variables of the nonlinear system. Unlike the traditional domain adaptation problem, the feature space in the target domain is different from that in the source domain. Therefore, the feature transformation is first performed. To simplify the heterogeneous domain adaptation problem, an assumption is first made as follows.

*Assumption* 1. The number of features in the source domain is the same as that in the target domain, that is, $d_{xs} = d_{xt} = d_x$ and $d_{ys} = d_{yt} = d_y$.

*Remark* 1. Assumption 1 can be easily satisfied in practice, when the input and output dimensions differ between the source and target domains. In such cases, feature augmentation techniques can be employed to align the number of features across both domains, such as repeating features or adding zero vectors. Note that although such alignment enables the pre-trained model to operate on the target domain, its performance is usually quite poor due to feature mismatch (Daumé III, 2009). Therefore, feature transformation is still needed to improve the performance of HTL methods.

We consider modeling the nonlinear system of Eq. 1 using neural networks. We define the set of hypothesis functions as $\mathcal{H} = \{h : (\mathbf{x}_1, \ldots, \mathbf{x}_t) \to \mathbf{y}_t\}$, where $(\mathbf{x}_1, \ldots, \mathbf{x}_t) \in \mathcal{R}^{d_x \times t}$ denotes the first $t$-time-step input and $\mathbf{y}_t \in \mathcal{R}^{d_y}$ is the output representing the prediction at the $t$th time step. The input to $h(\cdot)$ is the entire sequence of $t$ vectors, $(\mathbf{x}_1, \ldots, \mathbf{x}_t)$, which belongs to the space $R^{(d_x \times t)}$. This sequence-based input is essential for the task of modeling dynamical systems, as the prediction $\mathbf{y}_t$ depends on the history of the previous $t$ steps. The function $h$ therefore acts as a sequence-to-vector model (e.g., as implemented by an RNN). A new set of hypothesis functions can be defined as follows:

$$\mathcal{L}_{\mathcal{H}} = \{\mathcal{L} : \mathbf{x} \to \mathcal{L}(h(\mathbf{x}), f_{\mathcal{S}}(\mathbf{x})), h \in \mathcal{H}\} \tag{3}$$

where $f_{\mathcal{S}}$ denotes the label function of the samples in the source distribution $\mathcal{S}$. The loss function $\mathcal{L}$ is assumed to be a local Lipschitz continuous function associated with the hypothesis function $h \in \mathcal{H}$ that maps $\mathbf{x} \in \mathcal{R}^{d_x}$ to $[0, 1]$, and the Lipschitz constant is $\mathcal{L}_c$. The hypothesis function $h \in \mathcal{H}$ is selected to model the dynamic behavior in the source domain $\mathcal{S}$. Traditional, $h$ developed from the source domain can be applied to the target domain directly in the case of a homogeneous domain adaptation problem. However, in this work, the feature spaces in the source and target domains are different, indicating that a feature transformation is required for domain adaptation.

Inspired by the parameter tuning methods (Bao et al., 2023), the feature transformation scheme between the source and target domains is designed as follows:

$$X_S = P X_T, Y_T = Q Y_S \tag{4}$$

where $P \in \mathcal{R}^{d_x \times d_x}$ and $Q \in \mathcal{R}^{d_y \times d_y}$ are two matrices for feature transformation between source and target domains. Unlike the data-based transformation scheme, Eq. 4 is designed to adapt the pre-trained model $h$ to the target domain; such adaptation can be described as follows:

$$Y_S = h(X_S), Y_T = Q h(P X_T) = h^*(X_T) \tag{5}$$

where $h^*(\cdot) = Q h(P x)$. A new hypothesis function set can be defined as $\mathcal{H}^*$, where $h^* \in \mathcal{H}^*$. $\mathcal{H}^*$ is constructed via transformation of the hypothesis function set $\mathcal{H}$ under Eq. 5. Therefore, the hypothesis function $h \in \mathcal{H}$ shows the map between the input sample to the output sample for the source domain, and $h^* \in \mathcal{H}^*$ shows the map for the target domain. Similar to Eq. 3, a hypothesis function set can be defined as $\mathcal{L}_{\mathcal{H}^*}$ based on $\mathcal{H}^*$ with Lipschitz constant $\mathcal{L}_c^*$.

By selecting proper transformation matrices $P$ and $Q$, the mismatched features can be addressed and the performance of the pre-trained model on target domain can be improved. Despite the successful application of the parameter tuning method in the large language model, the selection of proper $P$ and $Q$ in the case with limited data is still under investigation, which is subject to the feature mismatch and the discrepancy between the source and target domains. This problem is particularly critical for modeling nonlinear systems with complex dynamics and strong interconnection. To address this problem, we will conduct a theoretical analysis for the generalization performance of $h^* \in \mathcal{H}^*$ in the target domain using statistical machine learning theory, in order to provide a theoretical guarantee for parameter tuning methods. To simplify the notation, we consider a pre-trained recurrent neural network (RNN) with a single hidden layer for the theoretical analysis. This model satisfies the conditions specified in Golowich et al. (2020), including bounded weights and a Lipschitz continuous activation function. It is important to note that the results can be extended to deeper RNNs and a broader class of network architectures.

## 5 THEORETICAL ANALYSIS

In this section, the generalization error bound for heterogeneous domain adaptation is investigated and the result will be used in the next section to design and guide the HTL learning process. Specifically, we analyze the performance of a hypothesis function $h$ in the target domain $\mathcal{T}$ under adaptation schemed in Eq. 5 with limited data, while $h$ is obtained through the data set collected from the source domain $\mathcal{S}$. The empirical source dataset $\hat{\mathcal{S}}$ and target dataset $\hat{\mathcal{T}}$ are made up of samples drawn independently and identically distributed (i.i.d.) from the source and target domains, respectively. Empirical Rademacher complexity (ERC) theory is commonly used to measure the richness of the hypothesis function set, and its definition is shown as follows.

*Definition* 1. (Mohri et al., 2018) Given a hypothesis class $\mathcal{F}$ of real-valued functions $f(\cdot)$, and a set of training data samples $S = (s_1, \ldots, s_m)$, the ERC of $\mathcal{F}$ is defined as:

$$\widehat{\Re}_S(\mathcal{F}) = \mathbb{E}_{\epsilon}\left[\sup_{f \in \mathcal{F}} \frac{1}{m}\left(\sum_{i=1}^{m} \epsilon_i f(s_i)\right)\right] \tag{6}$$

where $\epsilon = (\epsilon_1, \ldots, \epsilon_m)$ is a set of independent and identically distributed (i.i.d.) Rademacher random variables satisfying $\mathbb{P}(\epsilon_i = -1) = \mathbb{P}(\epsilon_i = 1) = 0.5$.

Unlike the domain adaptation problem with homogeneous feature spaces, the feature transformation scheme in Eq. 4 is implemented to adapt the pre-trained model $h$ to the target domain, and a new hypothesis function set $\mathcal{H}^*$ is constructed. First, we derive the ERC for the new hypothesis function set $\mathcal{H}^*$ on the target dataset.

*Lemma* 1. The ERC of the hypothesis function set $\mathcal{H}^*$ with respect to the target dataset $\hat{\mathcal{T}}$ with $m$ samples satisfies the following inequality:

$$\widehat{\Re}_{\mathcal{T}}(\mathcal{L}_{\mathcal{H}^*}) \leq \sqrt{2} L_c^* d_y^2 Q_m \frac{M(\sqrt{2\log(2)t} + 1)P^m B_{XT}}{\sqrt{m}} \tag{7}$$

where $Q^m = \max_{i,j}|Q_{i,j}|$, $M = B_{V,F}B_{W,F}\frac{(B_{U,F})^t - 1}{B_{U,F} - 1}$, $P^m = \max_{i,j}|P_{i,j}|$, and $B_{XT}$ is the upper bound for $\mathbf{x_i} \in \hat{\mathcal{T}}$ such that $|\mathbf{x_i}| \leq B_{XT}$. $B_{V,F}$, $B_{W,F}$, and $B_{U,F}$ denote the upper bounds of the weight parameters in the neural network model $h$, specifically for the output layer ($B_{V,F}$) and the hidden layer (input $B_{W,F}$ and hidden state $B_{U,F}$), respectively. Similarly, we can denote the upper bound for $\mathbf{x_i} \in \hat{\mathcal{S}}$ as $|\mathbf{x_i}| \leq B_{XS}$.

The proof of Lemma 1 can be found in Appendix A.1. Lemma 1 measures the complexity of the hypothesis function set $\mathcal{H}^*$ on the target domain, which is related to the weight parameters in the pre-trained model, and the values in the transformation matrices. Lemma 1 will be used later to derive the generalization error bound, and guide the design of transformation matrices.

As a powerful tool in analyzing the domain adaptation problem, $\mathcal{Y}$-Discrepancy distance can effectively measure the similarity between two datasets, of which the definition is described as follows.

*Definition* 2. (Mohri & Muñoz Medina, 2012) Given a loss function $L(\cdot, \cdot) : \mathbf{Y} \times \mathbf{Y} \rightarrow \mathbf{R}_+$ associated with the hypothesis functions set $\mathcal{H}$, mapping $\mathbf{X}$ to $\mathbf{Y}$. The $\mathcal{Y}$-Discrepancy distance $disc_{YH}$ between two distributions $D_1$ and $D_2$ is defined as:

$$disc_{YH}(D_1, D_2) := \sup_{h \in \mathcal{H}} |\mathcal{L}_{D_1}(h, f_{D_1}) - \mathcal{L}_{D_2}(h, f_{D_2})| \tag{8}$$

where $\mathcal{L}_{D_1}(h, f_{D_1}) := \mathbb{E}_{\mathbf{x} \sim D_1}[L(h(\mathbf{x}), f_{D_1}(\mathbf{x}))]$. $f_{D_1}$ and $f_{D_2}$ denote labeled functions for the distributions $D_1$ and $D_2$, respectively.

Based on the definition of the $\mathcal{Y}$-Discrepancy distance, we can obtain the following inequality, which measures the relationship between the generalization error of the pre-trained model $h$ on the source domain $\mathcal{S}$ and its performance on the target domain $\mathcal{T}$ after feature transformation $h^*$.

$$\begin{aligned}\mathcal{L}_{\mathcal{T}}(h^*, f_{\mathcal{T}}) - \mathcal{L}_{\mathcal{S}}(h, f_{\mathcal{S}}) =& \mathcal{L}_{\mathcal{T}}(h^*, f_{\mathcal{T}}) - \mathcal{L}_{\mathcal{S}}(h^*, f_{\mathcal{S}}) + \mathcal{L}_{\mathcal{S}}(h^*, f_{\mathcal{S}}) - \mathcal{L}_{\mathcal{S}}(h, f_{\mathcal{S}}) \\ \leq& disc_{YH^*}(\mathcal{S}, \mathcal{T}) + \mathbb{E}_{x \sim \mathcal{S}}[L(h^*, f_{\mathcal{S}}) - L(h, f_{\mathcal{S}})]\end{aligned} \tag{9}$$

Then, the generalization error bound of the pre-trained model after feature transformation $h^*$ on the target dataset $\mathcal{T}$ can be derived in the following theorem.

**Theorem 1.** *Let $\mathcal{H}$ be a set of the hypotheses $h(\cdot)$ that map the RNN input to the $t$-th RNN output, and construct the hypothesis function set $\mathcal{H}^*$ based on Eq. 5. Given a dataset $\hat{\mathcal{S}}$ of $m_s$ i.i.d. samples from the source domain $\mathcal{S}$, and a dataset $\hat{\mathcal{T}}$ of $m_t$ i.i.d. data samples from the target domain $\mathcal{T}$, the following inequality holds with probability at least $1 - \delta$ for any hypothesis function $h^* \in \mathcal{H}^*$:*

$$\begin{aligned}\mathbb{E}_{x \sim \mathcal{T}}[L(h^*, f_{\mathcal{T}})] \leq& disc_{YH^*}(\hat{\mathcal{S}}, \hat{\mathcal{T}}) + L_c^* \mathbb{E}_{x \sim \mathcal{S}}[|h^* - h|] + \mathcal{L}_{\hat{\mathcal{S}}}(h, f_{\mathcal{S}}) + 6\sqrt{\frac{\log\left(\frac{6}{\delta}\right)}{2m_s}} + 3\sqrt{\frac{\log\left(\frac{6}{\delta}\right)}{2m_t}} \\ &+ 2\sqrt{2}\left(\sqrt{2\log(2)t} + 1\right)Md_y^2\left(L_c^* Q^m P^m\left(\frac{B_{XS}}{\sqrt{m_s}} + \frac{B_{XT}}{\sqrt{m_t}}\right) + L_c\frac{B_{XS}}{\sqrt{m_s}}\right)\end{aligned} \tag{10}$$

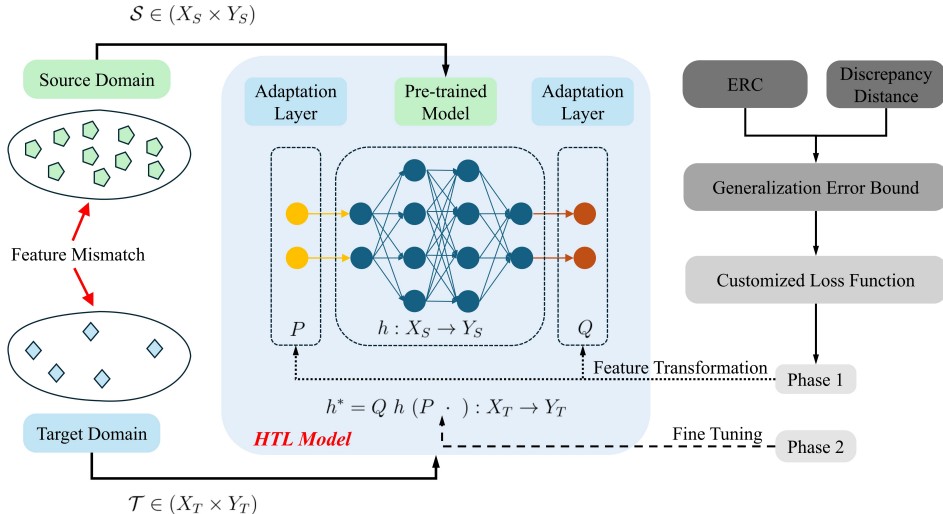

Figure 1: Framework of the proposed heterogeneous transfer learning algorithm.

The proof of Theorem 1 can be found in Appendix A.2. The first term in Eq. 10 $disc_{YH^*}(\hat{\mathcal{S}}, \hat{\mathcal{T}}) :=$ $\sup_{h \in \mathcal{H}^*} \left| \mathcal{L}_{\hat{\mathcal{S}}}(h, f_{\mathcal{S}}) - \mathcal{L}_{\hat{\mathcal{T}}}(h, f_{\mathcal{T}}) \right|$ denotes the $\mathcal{Y}$-discrepancy distance of the corresponding source dataset $\hat{\mathcal{S}}$ and corresponding target dataset $\hat{\mathcal{S}}$ over the hypothesis function set $\mathcal{L}_{\mathcal{H}^*}$. The second term denotes the difference between the original hypothesis function $h$ and the one after the feature transformation $h^*$ on the source dataset. The third term in the upper bound is the prediction performance of $h$ over the corresponding source dataset $\hat{\mathcal{S}}$. The fourth and fifth terms are the terms of probabilities. The last term denotes the sum of the ERC of $\mathcal{L}_{\mathcal{H}^*}$ over $\hat{\mathcal{S}}$ and $\hat{\mathcal{T}}$, and the ERC of $\mathcal{L}_{\mathcal{H}}$ over $\hat{\mathcal{S}}$. Theorem 1 can provide a guideline for improving the performance of the pre-trained model on the target domain with feature transformation, which is important for heterogeneous transfer learning algorithms. Specifically, the generalization error bound can be reduced in several ways, such as increasing the number of source and target data samples and optimizing feature transformation. In the next section, we will present the design of the heterogeneous transfer learning framework, guided by the analysis of Theorem 1, with a focus on the optimization of the transformation matrices $P$ and $Q$.

*Remark* 2. The necessity of Assumption 1 is strictly for our theoretical analysis, specifically to derive the generalization error bound in Theorem 1. This bound requires calculating the performance discrepancy between the original model ($h$) and the transformed model ($h^*$), which is only tractable if both models share the same input/output dimensions. We acknowledge that this is a theoretical constraint. In practice, the source and target domains may not share the same number of features. However, as shown in Remark 1, this assumption can be easily satisfied in practice using standard feature augmentation methods (e.g., zero-padding) to equalize the feature dimensions before applying the transformations.

## 6 METHODOLOGY

The objective of this work is to design a heterogeneous transfer learning framework for cases where the source and target domains have different feature spaces. Specifically, the feature transformation is first performed as shown in Eq. 4, and the generalization error bound for heterogeneous domain adaptation is derived in Theorem 1. Based on this theoretical result, the HTL algorithm is designed to enable effective knowledge transfer from the source to the target domain.

In this work, adaptation layers are added to the pre-trained model and serve as the feature transformation matrices, whose parameters are updated under the designed loss function. Specifically, we design a neural network layer as the adaptation layer, where the activation function is selected as the linear function, and the bias parameters are set as $0$. The explicit output of the adaptation layer with $k$ neurons can be described as $W\mu \in \mathcal{R}^k$, where $\mu \in \mathcal{R}^{d_\mu}$ is the input vector, and $W \in \mathcal{R}^{k \times d_\mu}$

denotes the weight parameters in the adaptation layer. By carefully selecting the number of neurons in the adaptation layer, the feature transformation matrices can be realized. The neural network structure for the heterogeneous transfer learning model is shown in Fig. 1. Specifically, an adaptation layer with $d_x$ neurons is added before the input layer of the pre-trained model, with weight parameter vector $W_{in} \in \mathcal{R}^{d_x \times d_x}$, which performs the input feature transformation as $P$. Similarly, an adaptation layer is added after the output layer of the pre-trained model with $d_y$ neurons, and the weight matrix $W_{out} = Q \in \mathcal{R}^{d_y \times d_y}$. By adjusting the neural network structure, the adaptation layer can be utilized as the feature transformation matrices, whose parameters are updated with the "ADAM optimizer", a common optimizer for updating neural network weights via gradient descent.

To identify the optimal feature transformation matrices, we design an optimization problem informed by the theoretical generalization error analysis in Theorem 1, in addition to modifying the neural network architecture. Specifically, our goal is to select feature transformation matrices that yield the best generalization performance of the pre-trained model on the target domain. As shown by Theorem 1, the generalization performance depends on multiple terms, including the discrepancy between the empirical source and target datasets, the prediction error between the original pre-trained model $h$ and the model after feature transformation $h^*$, the performance of the pre-trained model on the source domain, the probability-related term, and the weights of both the pre-trained model and transformation matrices. However, Theorem 1 cannot be directly used as the objective function for optimizing the transformation matrices, as many of its terms are difficult to compute. Moreover, using it directly in training would complicate the gradient computation, making it inefficient. Therefore, instead of designing the objective function exactly using the theoretical bound, we incorporate several key terms that strongly influence the generalization performance and are closely related to the transformation matrices. Specifically, an optimization problem is constructed for learning the feature transformation matrices, where the objective function is designed as:

$$Loss = \lambda_0 \mathcal{L}_{\hat{\mathcal{T}}}\left(h^*, f_{\hat{\mathcal{T}}}\right) + \lambda_1 \left|\mathcal{L}_{\hat{\mathcal{S}}}\left(h^*, f_{\hat{\mathcal{S}}}\right) - \mathcal{L}_{\hat{\mathcal{T}}}\left(h^*, f_{\hat{\mathcal{T}}}\right)\right| + \lambda_2 \mathbb{E}_{x \sim \hat{\mathcal{S}}}[||h^* - h||] + \lambda_3 Q_m P_m \quad (11)$$

where $\lambda_0$, $\lambda_1$, $\lambda_2$, and $\lambda_3$ denote the weight coefficients for different terms in the objective function. Specifically, the first term in Eq. 11 is the MSE between the predicted value and the ground truth for $h^*$ on the empirical training dataset collected from the target domain, which is a common loss function for regression tasks. The second term measures the performance difference of $h^*$ on the empirical source and target domains. The third term is inspired by the second term in Theorem 1. The last term represents the sum of ERCs, with a focus on the effect of the feature transformation matrices.

To obtain the heterogeneous transfer learning model, a two-phase training procedure is designed, as shown in Algorithm 1. Specifically, in Phase 1, we aim to identify the optimal feature transformation matrices $P$ and $Q$ (i.e., the weights in the adaptation layers), while the pre-trained model $h$ is fixed, achieved by freezing the weight parameters in the pre-trained model. The optimization process is implemented by training the adaptation layers under the customized loss function in Eq. 11 with the limited target data. After obtaining the feature transformation matrices that adapt the pre-trained model from the source domain to the target domain with good generalization performance, all parameters are set as trainable, and a fine-tuning method is utilized to update all weight parameters in the HTL model under the MSE loss function.

The implementation of matrices $P$ and $Q$ via customized adaptation layers has multiple benefits. It is not only efficient and straightforward to implement but also provides a strong initial estimate of model parameters in the target domain. Moreover, the training of the adaptation layers is guided by theoretical analysis, which ensures that the transformed features align well with the generalization objectives. This theoretically grounded guidance enhances the model's ability to generalize across different domains and tasks. In addition, because the adaptation layers are lightweight and modular, they can be easily integrated into various network architectures, making the proposed method applicable and scalable to a wide range of applications.

---

**Algorithm 1** Two-Phase Heterogeneous Transfer Learning Algorithm

---

**Require:** Pre-trained source model $h(\cdot; \theta_h)$, source dataset $\mathcal{D}_{\text{source}}$, target dataset $\mathcal{D}_{\text{target}}$, transformation layers $P(\cdot; \theta_P)$ and $Q(\cdot; \theta_Q)$ implemented as customized neural network layers.

 1: **Phase 1: Feature Space Adaptation**
 2: Construct the composite target model $h^*(x) = Q(h(P(x)))$.
 3: Update transformation parameters $(\theta_P, \theta_Q)$ by minimizing loss function in Eq. 11 on $\mathcal{D}_{\text{target}}$ and $\mathcal{D}_{\text{source}}$ while holding the source parameters $\theta_h$ fixed.
 4: **Phase 2: Full Network Fine-tuning**
 5: Update all parameters $\{\theta_h, \theta_P, \theta_Q\}$ by minimizing MSE loss on the target dataset $\mathcal{D}_{\text{target}}$.
 6: **return** Obtain the model $h^*$ on the target domain with updated parameters $\{\theta_h, \theta_P, \theta_Q\}$.

---

## 7 EXPERIMENTS

In this section, four experimental cases are used to demonstrate the effectiveness of the proposed method, ranging from standard dynamical systems to a complex, chaotic weather prediction task. The baselines are defined as follows. 1) From-scratch (FC): A randomly initialized model with the same architecture as the pre-trained model is trained from scratch using only the target data. 2) Fine-tuning (FT): The pre-trained model is fine-tuned directly on the target dataset, which is a standard transfer learning method (Jiang et al., 2022). 3) Adapter-tuning (AT): Adapter layers are inserted into the pre-trained model and trained first, followed by fine-tuning the entire network (Jiang et al., 2022). 4) LEADS: The model is decomposed into a shared component and an environment-specific component, which are then optimized with a custom loss function (Yin et al., 2021). The feature alignment method is first applied by repeating features to ensure that the feature dimensions are the same for the source and target domains, thus satisfying Assumption 1. All other settings, including data sets, the number of training epochs, and the neural network layers, are kept the same for all experiments. The training and testing errors (MSE) of the five methods for modeling the target process are presented in Fig. 2. All training processes are conducted on an Intel Core i7 with 32 GB of RAM. The details of the experiments are provided in Appendix B.

To strictly validate the proposed architecture, we conduct comprehensive ablation studies analyzing the specific contribution of each training stage. Furthermore, we evaluate the robustness of the proposed method under challenging conditions, including limited data, high-noise, and different feature augmentation strategies. These robustness analyses, detailed in Appendix C, demonstrate the resilience of our method compared to baselines, highlighting the effectiveness of the designed heterogeneous feature transformation matrices.

### 7.1 CHEMICAL REACTOR

We begin by applying the proposed method and the four benchmark models to the modeling of a continuous stirred tank reactor (CSTR), which is a commonly used nonlinear system in chemical engineering. The modeling task is to predict the state trajectory of the reactor over a time period based on its current state and the manipulated input during that period. The target process is the CSTR described in Abdullah et al. (2022), where the reaction is carried out to produce product $A$ from reactants $B$ and $C$. The CSTR in Alanqar et al. (2015) is selected as the source process, where a different reaction converts reactant $A$ to product $B$. The datasets are generated by simulating the first-principles models for the source and target processes. Limited data is collected for the target process, while sufficient data is collected from the source process to develop a high-accuracy model in the source domain. However, the number of states and their physical meanings used to model the source and target processes differ, which makes knowledge transfer from the source to the target process quite challenging. As shown in Fig. 2(a), our proposed HTL algorithm outperforms all benchmarks, achieving the smallest testing error ($2.5 \times 10^{-3}$) and the smallest discrepancy ($3 \times 10^{-4}$) between training and testing errors, demonstrating not only good prediction accuracy but also improved generalization on the target domain. Note that the pre-trained model performs poorly on the target domain due to feature mismatch, with a prediction MSE of 2.0, which is even higher than that of the fully-connected model FC with randomly initialized weights (1.7). However, the final HTL model significantly outperforms the fully connected model FC (with a error of $6.3 \times 10^{-3}$), showing the effectiveness of the proposed HTL methods in utilizing knowledge from the pre-trained

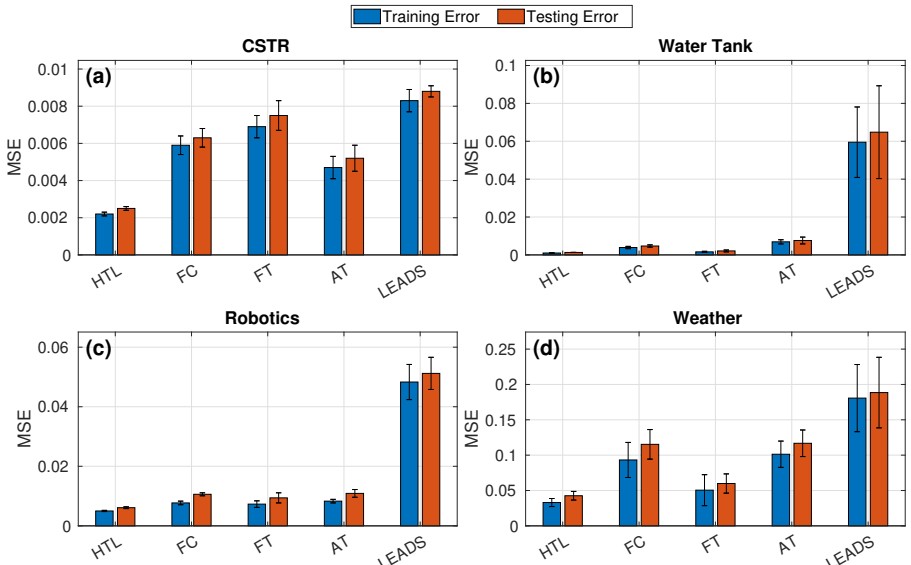

Figure 2: Training and testing error (MSE) of five methods for modeling the chemical reactor process (a), cascaded water tank (b), robotic system (c), and weather prediction (d).

model. The LEADS method yields a suboptimal testing error of $8.8 \times 10^{-3}$, largely due to its restrictive additive assumption ($f = f_{share} + f_{specific}$). While our domains share underlying dynamics, this simple decomposition cannot capture the more complex interactions between shared $f_{share}$ and domain-specific $f_{specific}$ components, resulting in poor performance.

## 7.2 CASCADED WATER TANK

In this case study, the real-world cascaded tank system from Schoukens et al. (2016) is selected as the target process, which consists of two water tanks and is subjected to disturbances and noise. The source process is selected as the first-principles model of the water tank process. Specifically, the state variables for the system are the water levels in two tanks, while the output variable denotes the water level in the lower tank, which is subject to noise. The features of the source process include the input and state vectors, whereas the target process consists of the input and output vectors. The results shown in Fig. 2(b) illustrate the effectiveness of the proposed HTL methods, achieving the lowest testing MSE for modeling real nonlinear systems under complex conditions by effectively leveraging the pre-trained model developed from simulation data of the first-principles model.

## 7.3 ROBOTIC SYSTEM

In this case study, a three-degree-of-freedom torque-controlled robotic system from Agudelo-España et al. (2020) is used, which includes three joints, each with three state variables and one input variable. The state trajectories collected from the simulation and real experiments of the closed-loop system are used as the source and target domains, respectively. Specifically, the data collected for the first and second joints are used as the source process, while the target data includes all three joints. The number of input and output features in the target domain is 12 and 9, respectively. To address the feature mismatch, we first repeat the data for the second joint in the source domain to align the number of features with that of the target domain. Then, a pre-trained model is developed using the augmented source dataset. As shown in Fig. 2(c), the proposed HTL method outperforms all benchmarks, demonstrating the effectiveness of our approach on the high-dimensional nonlinear system. Specifically, the testing error of the HTL model is $6.1 \times 10^{-3}$, which significantly lower than that of the standard fully-connected model FC ($1.1 \times 10^{-2}$) that does not utilize the pre-trained model or the neural network structure changes guided by the theoretical analysis. Moreover, it also outperforms the FT, AT, and LEADS, further validating the advantages of the proposed framework.

### 7.4 WEATHER PREDICTION

A new, challenging case study on a chaotic, real-world dynamical system (i.e., weather prediction) is considered. The objective is to transfer a model from New York (JFK) (source) to Los Angeles (LAX) (target). This task is a rigorous test due to heterogeneity and data scarcity. The dataset is obtained via Meteostat (https://dev.meteostat.net/). Specifically, the source (JFK) lacks pressure (P) data, resulting in different input and output spaces from the target (LAX). The target has only 219 training samples to be validated against a large (43,603) test set. As shown in Fig. 2(d), despite the system's chaotic nature and the severe data limitation, our HTL method achieves the lowest test MSE (0.0426). It performs 1.4 times better than the next-best baseline, FT (0.0599), and significantly outperforms AT (0.1168). This result strongly demonstrates that our HTL framework is not limited to one class of problems and can effectively transfer knowledge to complex domains, even under high heterogeneity and data scarcity.

### 7.5 ABLATION STUDIES

To illustrate the necessity of each major component, four ablation experiments are designed: Case AA (Without Pre-training), Case AB (Without Feature Adaptation), Case AC (Without Fine-tuning), Case AD (Without Fine-tuning, Extended Adaptation). The results for the ablation studies and the HTL method are summarized in the following table.

Table 1: Testing errors for ablation cases (Mean $\pm$ Std. All values scaled by $10^{-3}$)

|  | HTL | AA | AB | AC | AD |
|---|---|---|---|---|---|
| **CSTR** | $\mathbf{2.47 \pm 0.15}$ | $3.27 \pm 0.40$ | $8.20 \pm 0.96$ | $5.37 \pm 0.15$ | $3.90 \pm 0.00$ |
| **Water Tank** | $\mathbf{1.33 \pm 0.06}$ | $1.97 \pm 0.38$ | $2.30 \pm 0.69$ | $12.00 \pm 0.00$ | $2.30 \pm 0.00$ |
| **Robotic** | $\mathbf{6.13 \pm 0.32}$ | $7.20 \pm 0.00$ | $8.33 \pm 1.12$ | $52.33 \pm 9.02$ | $18.33 \pm 1.53$ |
| **Weather** | $\mathbf{42.6 \pm 6.1}$ | $77.5 \pm 9.2$ | $69.2 \pm 13.7$ | $112.3 \pm 45.8$ | $60.0 \pm 11.4$ |

The ablation study clearly demonstrates the effectiveness of each major component in our proposed framework. In all cases, removing any stage leads to substantial performance degradation. More critically, removing the feature adaptation stage causes severe performance degradation and even model collapse, highlighting that this stage is the core of our method. In summary, the pre-trained model provides a solid foundation, the adaptation stage performs the critical transformation, and the fine-tuning stage refines the model for optimal performance. The removal of any stage leads to a marked performance drop, confirming the well-justified design of our framework.

## 8 CONCLUSION

In this work, a novel heterogeneous transfer learning framework was proposed to address the feature mismatch between the source and target domains. Feature transformation matrices were implemented through customized adaptation layers that adapted the pre-trained model to the target domain. Theoretical analysis was performed to derive the generalization error bound for heterogeneous domain adaptation, which was utilized to guide the design of the loss function to train the adaptation layers. Finally, a fine-tuning step was applied to update all weight parameters in the HTL model. The effectiveness of the proposed method was demonstrated through four experimental case studies including a chemical reactor, a water tank system, a robotic platform, and a weather prediction task.

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

## REPRODUCIBILITY STATEMENT AND LLM USAGE STATEMENT

We provide implementation code, experimental scripts, dataset descriptions, and complete theoretical analyses and proofs in the supplementary materials to ensure reproducibility. We used a large language model (LLM) for proofreading and polishing. All authors reviewed and edited the content to ensure scientific accuracy.

## A THEORETICAL PROOF DETAILS

### A.1 PROOF OF LEMMA 1

The proof of Lemma 1 is shown as follows.

*Proof.* The Empirical Rademacher complexity (ERC) of the newly defined hypothesis function set $\mathcal{H}^*$ can be obtained as follows:

$$
\begin{aligned}
\widehat{\mathfrak{R}}_{\mathcal{T}}(\mathcal{L}_{\mathcal{H}^*}) &= \frac{1}{m} \mathbb{E}_{\sigma} \left[ \sup_{h^* \in \mathcal{H}^*} \left( \sum_{i=1}^{m} \sigma_i \mathcal{L}(h^*(\mathbf{x_i}), f_{\mathcal{T}}(\mathbf{x_i})) \right) \right] \\
&\leq \frac{\sqrt{2} L_c^*}{m} \mathbb{E}_{\sigma} \left[ \sup_{h^* \in \mathcal{H}^*} \left( \sum_{i=1}^{m} \sum_{k=1}^{d_y} \sigma_i h_k^*(\mathbf{x_i}) \right) \right] \\
&= \frac{\sqrt{2} L_c^*}{m} \mathbb{E}_{\sigma} \left[ \sup_{h \in \mathcal{H}} \left( \sum_{i=1}^{m} \sum_{k=1}^{d_y} \sum_{j=1}^{d_y} \sigma_i Q_{k,j} h_j(P\mathbf{x_i}) \right) \right] \\
&\leq \frac{\sqrt{2} L_c^* d_y Q_m}{m} \mathbb{E}_{\sigma} \left[ \sup_{h \in \mathcal{H}} \left( \sum_{i=1}^{m} \sum_{j=1}^{d_y} \sigma_i h_j(P\mathbf{x_i}) \right) \right] \\
&\leq \frac{\sqrt{2} L_c^* d_y Q_m}{m} \sum_{j=1}^{d_y} \mathbb{E}_{\sigma} \left[ \sup_{h \in \mathcal{H}_k} \left( \sum_{i=1}^{m} \sigma_i h(P\mathbf{x_i}) \right) \right] \\
&\leq \sqrt{2} L_c^* d_y Q_m \sum_{j=1}^{d_y} \widehat{\mathfrak{R}}_{\mathcal{T}^T}(\mathcal{H}_k) \\
&\leq \sqrt{2} L_c^* d_y^2 Q_m \frac{M(\sqrt{2\log(2)t} + 1) P^m B_{XT}}{\sqrt{m}}
\end{aligned}
\tag{12}
$$

The first line is obtained based on the definitions of the ERC and $\mathcal{H}^*$. The second line is derived following Corollary 4 in (Maurer, 2016). The third line is obtained based on the definition of $h^*$ in Eq. 5, and the fourth line is derived by defining $Q^m = \max_{i,j} |Q_{i,j}|$. By constructing a new dataset as $\mathcal{T}^T = (PX_T, Y_T)$ from $\mathcal{T}$, the fifth line is obtained. Note that samples in the newly constructed dataset $\mathcal{T}^T$ are i.i.d. The sixth line is obtained via the Ledoux-Talagrand inequality (Ledoux & Talagrand, 2013). The seventh inequality follows (Golowich et al., 2020), where $M = B_{V,F} B_{W,F} \frac{(B_{U,F})^t - 1}{B_{U,F} - 1}$, $P^m = \max_{i,j} |P_{i,j}|$, and $B_{XT}$ is the upper bound for $\mathbf{x_i} \in \hat{\mathcal{T}}$ such that $|\mathbf{x_i}| \leq B_{XT}$. Similarly, we can denote the upper bound for $\mathbf{x_i} \in \hat{\mathcal{S}}$ as $|\mathbf{x_i}| \leq B_{XS}$. $\qquad \square$

### A.2 PROOF OF THEOREM 1

*Proof.* Based on the definition of the Empirical Rademacher complexity, the following inequality holds with probability no less than $1 - \delta$ over the hypothesis function set $\mathcal{L}_{\mathcal{H}^*}$ and dataset $\mathcal{S}$:

$$
\text{disc}_{YH^*}(\mathcal{S}, \hat{\mathcal{S}}) = \left| \mathcal{L}_{\hat{\mathcal{S}}}(h, f_{\mathcal{S}}) - \mathcal{L}_{\mathcal{S}}(h, f_{\mathcal{S}}) \right| \leq 2\widehat{\mathfrak{R}}_S(L_{\mathcal{H}^*}) + 3\sqrt{\frac{\log\left(\frac{2}{\delta}\right)}{2m}}
\tag{13}
$$

Then, we have:

$$
\begin{aligned}
\mathcal{L}_{\mathcal{T}}\left(h^*, f_{\mathcal{T}}\right) \leq & disc_{YH^*}(\mathcal{S}, \mathcal{T}) + \mathcal{L}_{\mathcal{S}}\left(h, f_{\mathcal{S}}\right) + \mathbb{E}_{x \sim \mathcal{S}}[L\left(h^*, f_S\right) - L\left(h, f_S\right)] \\
\leq & disc_{YH^*}(\hat{\mathcal{S}}, \hat{\mathcal{T}}) + disc_{YH^*}(\mathcal{S}, \hat{\mathcal{S}}) + disc_{YH^*}(\mathcal{T}, \hat{\mathcal{T}}) \\
& + L_c^* \mathbb{E}_{x \sim \mathcal{S}}[|h^* - h|] + \mathcal{L}_{\hat{\mathcal{S}}}\left(h, f_S\right) + disc_{YH}(\mathcal{S}, \hat{\mathcal{S}}) \\
\leq & disc_{YH^*}(\hat{\mathcal{S}}, \hat{\mathcal{T}}) + L_c^* \mathbb{E}_{x \sim \mathcal{S}}[|h^* - h|] + \mathcal{L}_{\hat{\mathcal{S}}}\left(h, f_S\right) \\
& + 2\widehat{\Re}_{\mathcal{S}}\left(L_{\mathcal{H}^*}\right) + 2\widehat{\Re}_{\mathcal{T}}\left(L_{\mathcal{H}^*}\right) + 2\widehat{\Re}_{\mathcal{S}}\left(L_{\mathcal{H}}\right) \\
& + 6\sqrt{\frac{\log\left(\frac{6}{\delta}\right)}{2m_s}} + 3\sqrt{\frac{\log\left(\frac{6}{\delta}\right)}{2m_t}}
\end{aligned}
\tag{14}
$$

The first line is obtained via Eq. 9. The second inequality is derived via the triangle inequality and the definition of $\mathcal{Y}$-Discrepancy distance. The third inequality holds with probability at least $1 - \delta$, which is derived following Eq. 13 and the union-bound inequality.

Similar to Eq. 12, the ERC for $\widehat{\Re}_{\mathcal{T}}\left(L_{\mathcal{H}^*}\right)$ and $\widehat{\Re}_{\mathcal{S}}\left(L_{\mathcal{H}}\right)$ can be obtained. Therefore, we have

$$
\begin{aligned}
& \widehat{\Re}_{\mathcal{S}}\left(L_{\mathcal{H}^*}\right) + \widehat{\Re}_{\mathcal{T}}\left(L_{\mathcal{H}^*}\right) + \widehat{\Re}_{\mathcal{S}}\left(L_{\mathcal{H}}\right) \\
& \leq \sqrt{2} d_y^2 M \left(\sqrt{2\log(2)t} + 1\right) \left(L_c^* Q^m P^m \left(\frac{B_{XS}}{\sqrt{m_s}} + \frac{B_{XT}}{\sqrt{m_t}}\right) + L_c \frac{B_{XS}}{\sqrt{m_s}}\right)
\end{aligned}
\tag{15}
$$

Finally, the proof is completed by applying Eq. 14 and Eq. 15. $\qquad\square$

### A.3 DISCUSSIONS ON $P$ AND $Q$

Eq. 4 does not assume subspace equivalence. Rather, it defines $P$ and $Q$ as the learnable transformation matrices that map between the spaces. Therefore, $P$ and $Q$ are not required to be nonsingular (invertible). Their function is to serve as the learnable components of our transformed model, $h^*(X_T) = Qh(PX_T)$, as defined in Eq. 5. Our theoretical analysis, including the derivation of the generalization bound in Theorem 1, does not depend on the invertibility of these matrices, only that they are learnable transformations of the appropriate dimensions.

### A.4 DISCUSSIONS ON NONLINEAR FEATURE TRANSFORMATION METHOD

The selection of the linear transformation method is motivated by two critical factors, one theoretical and one practical. Theoretical Tractability: The linear transformation is essential for our theoretical contribution, as it permits the tractable derivation of the generalization error bound presented in Theorem 1. A nonlinear transformation would make this analysis significantly more complex and is beyond the scope of this paper's core theoretical guarantees. Practical Robustness: Our framework is designed for heterogeneous transfer, where the target domain often has limited data. A complex nonlinear transformation would introduce a large number of new parameters, creating a very high risk of overfitting. The linear approach is a more robust choice, making it far more suitable for these practical, data-scarce scenarios.

## B EXPERIMENT DETAILS

### B.1 MODELING A CHEMICAL REACTOR

In this case study, we consider a well-mixed, non-isothermal continuous stirred-tank reactor (CSTR) as the target process, in which the reaction that produces ethylbenzene (EB) from ethylene (E) and benzene (B) takes place. Specifically, the dynamics of the target process is described by the

following ordinary differential equations (ODEs) (Abdullah et al., 2022):

$$\frac{dC_A}{dt} = \frac{FC_{A0} - F_{out}C_A}{V} - ke^{\frac{-E}{RT}}C_AC_B$$

$$\frac{dC_B}{dt} = \frac{FC_{B0} - F_{out}C_B}{V} - ke^{\frac{-E}{RT}}C_AC_B \qquad (16)$$

$$\frac{dT}{dt} = \frac{FT_0 - F_{out}T}{V} - \frac{\Delta H}{\rho_L C_p}ke^{\frac{-E}{RT}}C_AC_B + \frac{Q}{\rho_L C_p V}$$

where $C_A$ and $C_B$ represent the concentrations of reactant A and product B, respectively, in the reactor, and $T$ is the reactor temperature. The input variable $Q$ denotes the heat supply rate. The parameters and steady-states of the nonlinear process can be found in (Abdullah et al., 2022). The integration time step is set to $0.01$ $hr$ to simulate the first-principles model in Eq. 16 using the explicit Euler method, and the sampling time is the same as the integration time. The objective is to model the nonlinear process in Eq. 16 using the proposed HTL method with limited data. Specifically, the input to the neural network (NN) model is $[C_A, C_B, T, Q]$, and the trajectories of the state variables $[C_A, C_B, T]$ over 5 sampling time steps are chosen as the outputs. A total of $10,000$ samples are collected from the target process, with $500$ training samples, and $9,500$ testing samples. This split method is designed to assess the generalization ability of the model when trained with limited data.

The standard fully-connected model (FC) developed for the target process consists of one hidden layer of 16 neurons under the $tanh(\cdot)$ activation function. The model is trained for 500 epochs using the mean squared error (MSE) between the predicted and true states as the loss function.

The dynamic behavior of the source process running a different reaction in the same type of reactor (i.e., CSTR) can be represented by the following ODEs:

$$\frac{dC_D}{dt} = \frac{FC_{D0} - F_{out}C_D}{V} - ke^{\frac{-E}{RT}}C_D^2$$

$$\frac{dT}{dt} = \frac{FT_0 - F_{out}T}{V} - \frac{\Delta H}{\rho_L C_p}ke^{\frac{-E}{RT}}C_D^2 + \frac{Q}{\rho_L C_p V} \qquad (17)$$

where $C_D$ denotes the concentration of reactant D, and $T$ is the reactor temperature. The input variables $C_{D0}$ and $Q$ denote the input flow rate of the reactant and the heat supply rate, respectively. The process parameters can be found in (Alanqar et al., 2015). Traditionally, the input vector for the neural network model used to learn the dynamics of the source process is $[C_D, T, C_{D0}, Q]$, while the output is the state trajectories of $[C_D, T]$. Since the number of output features for the source process is different from that of the target process, a feature alignment strategy is applied by repeating specific features. As a result, the output vector for the source process is augmented to $[C_D, C_D, T]$.

The pre-trained model developed for the source domain consists of one hidden layer of 16 neurons, and the activation function in the hidden layer is chosen as $tanh(\cdot)$. To ensure high prediction accuracy on the source process, $18,150$ training samples are collected, and the model is trained for $500$ epochs with the MSE loss function. The model achieves a testing error of $7.45 \times 10^{-4}$ on 6,050 test samples, illustrating desired prediction performance on the source process.

However, the pre-trained model performs poorly on the target domain due to the feature mismatch between the source and target domains. To develop the proposed HTL model, two adaptation layers are incorporated into the pre-trained model. Specifically, the adaptation layer corresponding to the matrix $P$ is implemented using three neural network layers with 8, 8, and 4 neurons, respectively. These layers have linear activation functions, and their bias terms are fixed to zero. Similarly, the adaptation layer for matrix $Q$ consists of three neural network layers with 4, 4, and 3 neurons.

The adaptation layer is implemented using multiple neural network layers to improve its performance for feature transformation. These layers can be combined by multiplying their weight matrices, thereby reducing them to a single equivalent neural network layer. For example, the weight matrices used in the layers for $P$ have dimensions $\mathcal{R}^{4\times8}, \mathcal{R}^{8\times8}, \mathcal{R}^{8\times4}$. By multiplying these matrices, a single transformation matrix of dimension $\mathcal{R}^{4\times4}$ can be obtained and used as a new adaptation layer. Specifically, the adaptation layers in the fine-tuning step are achieved via two single-layer neural networks with 3 and 2 neurons, corresponding to the feature transformation matrices $P$ and $Q$.

The parameters of these layers are initialized by multiplying the weight matrices obtained in the first phase.

The training phase of the proposed HTL method contains two stages. In the first phase, the pre-trained model is frozen, and only the weight parameters of the adaptation layers are updated using the target data. The loss function for this phase is designed in Eq. (11). After 100 epochs, the model is recompiled with all parameters set as trainable. In the second phase, the entire model is fine-tuned with 400 epochs using the MSE loss function.

In addition to the fully-connected models, two transfer learning models are developed as the baselines. The fine-tuning (FT) model is obtained via fine-tuning the pre-trained model directly using the target dataset for 500 epochs, and the loss function is chosen as MSE. The development of adapter-tuning (AT) leverages the pre-trained model with added adaptation layers and follows a two-phase training procedure totaling 500 epochs. Specifically, the output layer in the pre-trained model is removed , and a new hidden layer and a new output layer is added upon the pre-trained model as the adaptation layers. The first phase updates the adaptation layers, and the second phase fine-tunes all parameters.

To develop the LEADS model, three source processes are chosen, each described by Eq. 17 but with distinct parameters. The datasets collected from these processes represent different environments. The shared dynamics $f_{share}$ is captured via a neural network with the same structure as the pre-trained model, while three neural networks are developed to represent the domain-specific terms for the three sources. For each source process, $18, 150$ training samples are collected, and the LEADS model is first trained with the three source datasets under the designed loss function in Yin et al. (2021). To adapt the LEADS model to the target domain, $f_{share}$ is retained as the shared model, and a new domain-specific model is introduced. The adaptation follows a two-stage process with 500 epochs in total. In the first stage, the parameters of the domain-specific model are updated while keeping the shared dynamics fixed. In the second stage, all parameters are fine-tuned using the target dataset under the MSE loss function.

Each experiment is repeated three times. The prediction errors on both the training and testing datasets for all five methods are reported in Table 2, including the mean and standard deviation values.

Table 2: Model Performance Comparison for CSTR (Mean $\pm$ Std over 3 runs)

| Method | Train Error | Test Error |
|--------|-------------|------------|
| **HTL** | $0.0022 \pm 0.0001$ | $0.0025 \pm 0.0001$ |
| **FC** | $0.0059 \pm 0.0005$ | $0.0063 \pm 0.0005$ |
| **FT** | $0.0069 \pm 0.0006$ | $0.0075 \pm 0.0008$ |
| **AT** | $0.0047 \pm 0.0006$ | $0.0052 \pm 0.0007$ |
| **LEADS** | $0.0083 \pm 0.0006$ | $0.0088 \pm 0.0003$ |

### B.2 MODELING CASCADED WATER TANKS

The real-world cascaded water tank system in (Schoukens et al., 2016) is considered as the target process in this case study. The illustration of the system is shown in Fig. 3. The target process consists of two water tanks with a pump, where the water levels in the upper and lower tanks are chosen as state variables $h_1$ and $h_2$, respectively. The input variable $u$ is the voltage for the pump, which delivers the water into the upper tank from a reservoir. The water then flows through the lower tank and returns to the reservoir. The measured water level in the lower tank is designed as the output variable $y$, which is subject to measurement noise. The experimental dataset consists of input and output signals over 4096 seconds with 1024 data points sampled at 4-second intervals.

The objective is to model the nonlinear process using the experimental data via the proposed HTL method. Specifically, the goal is to predict the output signal $y(t + 1)$ based on the current output $y(t)$ and the manipulated input $u(t)$. The neural network model learns the input-output mapping: $[u(t), y(t)] \rightarrow y(t + 1)$. The dataset is divided into 204 training samples and 819 testing samples.

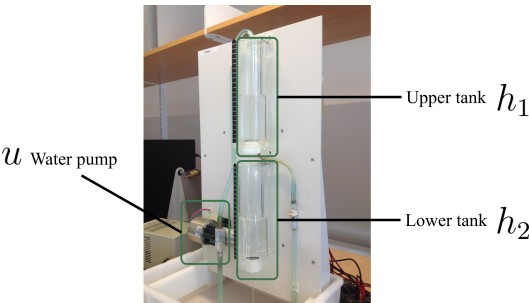

Figure 3: Illustration of cascaded water tank system (Schoukens et al., 2016).

The source process is selected as the first-principles model of the cascaded water tanks, derived using the Bernoulli's principles in (Schoukens et al., 2016). The system is described as:

$$
\begin{aligned}
\dot{h}_1(t) &= -k_1\sqrt{h_1(t)} + k_4 u(t) + d_1(t) \\
\dot{h}_2(t) &= k_2\sqrt{h_1(t)} - k_3\sqrt{h_2(t)} + d_2(t) \\
y(t) &= h_2(t) + w(t)
\end{aligned}
\tag{18}
$$

where $d_1$, $d_2$, and $w$ are noise terms. The system parameters $k_1$, $k_2$, $k_3$, and $k_4$ are all set to 1. The neural network developed to model the source process aims to predict the state variables, and is trained using the input-output mapping $[h_1(t), h_2(t), u(t)] \rightarrow [h_1(t), h_2(t)]$. Sufficient data is generated by simulating the first-principles model in Python, resulting in 12,000 training samples and 4,000 testing samples. The pre-trained model is with one hidden layer of 8 neurons and the $tanh(\cdot)$ activation function. The loss function is selected as MSE. After 500 training epochs, the model achieves a prediction MSE of $2.3 \times 10^{-5}$ on the testing set, indicating high accuracy in capturing the source process dynamics.

Since the number of input and output features differs between the source and target processes, a feature alignment method is first implemented for the target dataset by repeating the feature $y$. Therefore, the neural network model for the target process adopts the input-output mapping $[u(t), y(t), y(t)] \rightarrow [y(t+1), y(t+1)]$. Similar to modeling the CSTR in Appendix B.1, the fully-connected model $FC$ developed for the target process is with one hidden layer of 8 neurons and the $tanh(\cdot)$ activation function. FC is trained for 400 epochs under the MSE loss function.

The adaptation for matrix $P$ is achieved using three neural network layers with 8, 8, and 3 neurons, while the matrix $Q$ uses three neural network layers of 4, 4, and 2 neurons. In the first phase, the adaptation layers in the proposed HTL model are updated using the customized loss function for 100 epochs. Then, the multi-layer adaptation networks for $P$ and $Q$ are reduced to two single-layer networks with 3 and 2 neurons, respectively. Their parameters are initialized by multiplying the weight matrices from the corresponding multi-layer adaptation networks. The fine-tuning step for the entire model consists of 200 epochs.

FT is obtained by fine-tuning the pre-trained model directly for 300 epochs using the MSE loss function. AT is developed via removing the output layer and adding new hidden and output layers in the pre-trained model. It also follows the two-stage training process, with adaptation layers under 100 epochs and the entire model is then fine-tuned with 200 epochs.

To develop the LEADS model, three source processes are selected to denote three environments, which are all represented by Eq. 18 but with different parameters. The development and adaptation of the LEADS method follows the procedure outlined in Appendix B.1, except that the shared dynamics is represented via a neural network with the same structure as the pre-trained model (i.e., one hidden layer of 8 neurons), and the total number of epochs for the adaptation is 300.

The training and testing errors for all five methods are listed in Table 3, where each experiment is repeated three times.

Table 3: Model Performance Comparison for Water tank (Mean $\pm$ Std over 3 runs)

| Method | Train Error | Test Error |
|--------|-------------|------------|
| HTL | $0.0010 \pm 0.0000$ | $0.0013 \pm 0.0000$ |
| FC | $0.0039 \pm 0.0006$ | $0.0047 \pm 0.0006$ |
| FT | $0.0016 \pm 0.0003$ | $0.0021 \pm 0.0005$ |
| AT | $0.0069 \pm 0.0011$ | $0.0076 \pm 0.0018$ |
| LEADS | $0.0595 \pm 0.0186$ | $0.0648 \pm 0.0245$ |

### B.3 MODELING A ROBOTIC SYSTEM

A real-world robot dataset from (Agudelo-España et al., 2020) is utilized in this case study to demonstrate the effectiveness of the proposed method on a high-dimensional nonlinear system. The illustration of the three degree-of-freedom (three DOF) torque-controlled robotic system is shown in Fig. 4, which comprises three joints, and the state variables used to represent the robotic dynamics are the angles, velocities, and torques for the three joints (i.e., $[q_i, v_i, \tau_i]$ for the $i$-th joint, with $i = 1, 2, 3$). The input variables to the nonlinear system are the constrained control torques $\tau_{ui}$ applied to each joint.

The objective is to predict the state variables based on their current values and the manipulated inputs. Specifically, the neural network model is constructed to learn the mapping:

$$[q_1(t), v_1(t), \tau_1(t), q_2(t), v_2(t), \tau_2(t), q_3(t), v_3(t), \tau_3(t), \tau_{u1}(t), \tau_{u2}(t), \tau_{u3}(t)]$$
$$\rightarrow [q_1(t+1), v_1(t+1), \tau_1(t+1), q_2(t+1), v_2(t+1), \tau_2(t+1), q_3(t+1), v_3(t+1), \tau_3(t+1)] \tag{19}$$

State trajectories are collected from the physical robotic system. Specifically, a trajectory consisting of 14999 data points is selected as the target process, which is divided into 749 training samples and 14250 testing samples. The fully-connected model FC developed for the target process is with two hidden layers of 16 neurons each, using the $tanh(\cdot)$ activation function. The number of training epochs is 400, and the loss function is MSE.

The simulation data for the robotic platform is used to construct the source dataset. Specifically, we use simulated trajectories for the first and second joints, assuming that the simulation data for the third joint is unavailable. The feature mismatch between the source and target domains is addressed by repeating the features for the second joint. As a result, the pre-trained model is developed for the following input-output mapping:

$$[q_1(t), v_1(t), \tau_1(t), q_2(t), v_2(t), \tau_2(t), q_2(t), v_2(t), \tau_2(t), \tau_{u1}(t), \tau_{u2}(t), \tau_{u2}(t)]$$
$$\rightarrow [q_1(t+1), v_1(t+1), \tau_1(t+1), q_2(t+1), v_2(t+1), \tau_2(t+1), q_2(t+1), v_2(t+1), \tau_2(t+1)] \tag{20}$$

The pre-trained model is developed with two hidden layers of 16 neurons each; note that the structure of the pre-trained model is the same as FC. Around $70,000$ samples from the source dataset are utilized to train the pre-trained model under the MSE loss function for 400 epochs. The testing error on the source domain is $5.9 \times 10^{-3}$, suggesting good prediction performance.

Similar to Appendix B.1 and Appendix B.2, to develop the target model using the proposed HTL method, two adaptation layers are added to the pre-trained model. The adaptation layers are trained for 100 epochs under the customized loss function, followed by the fine-tuning step to update all weight parameters for 300 epochs with the MSE loss function. FT is obtained by fine-tuning the pre-trained model for 400 epochs under the MSE loss function. AT is developed via the adapter-tuning method. In LEADS method, the source process is selected as the simulation data with mismatched features. The performances of the five methods are listed in Table 4.

### B.4 WEATHER PREDICTION

We consider a new, challenging, cross-domain case study that moves beyond continuous control systems. We selected weather prediction, which involves modeling a complex, chaotic dynamical

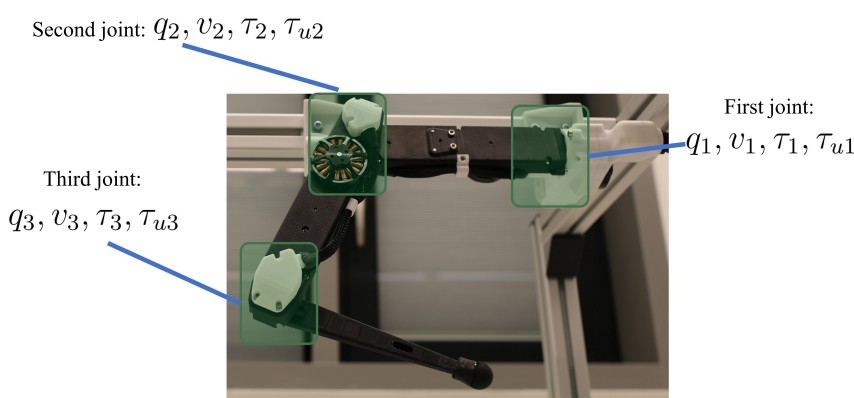

Figure 4: Illustration of a three degree-of-freedom torque-controlled robotic system (Agudelo-España et al., 2020).

Table 4: Model Performance Comparison for Robotics (Mean $\pm$ Std over 3 runs)

| Method | Train Error | Test Error |
|--------|-------------|------------|
| **HTL** | $0.0050 \pm 0.0002$ | $0.0061 \pm 0.0003$ |
| **FC** | $0.0077 \pm 0.0006$ | $0.0106 \pm 0.0005$ |
| **FT** | $0.0073 \pm 0.0011$ | $0.0094 \pm 0.0017$ |
| **AT** | $0.0083 \pm 0.0006$ | $0.0109 \pm 0.0013$ |
| **LEADS** | $0.0483 \pm 0.0059$ | $0.0512 \pm 0.0054$ |

system. The objective is to predict future temperature (T) and pressure (P) in Los Angeles (LAX) (the target domain) by leveraging a pre-trained model from New York (JFK) (the source domain). The source dataset (JFK) contains 87,646 samples, while the target dataset (LAX) is highly limited, with only 219 training samples and 43,603 test samples. The dataset is obtained via Meteostat (https://dev.meteostat.net/). This task is fundamentally heterogeneous. The source domain (JFK) lacks pressure data. Its input-output relation, using a 3-hour lookback, is

$$[T(t), T(t), W(t), \ldots, T(t+2), T(t+2), W(t+2)] \rightarrow [T(t+3), T(t+3)]$$

The target domain (LAX) uses all features as

$$[T(t), P(t), W(t), \ldots, T(t+2), P(t+2), W(t+2)] \rightarrow [T(t+3), P(t+3)]$$

The pre-trained model was developed with one hidden layer of 16 neurons. We compared four methods. HTL (Ours): The feature transformation stage was trained for 50 epochs, followed by 50 fine-tuning epochs. FT: The pre-trained model was fine-tuned for 100 epochs. FC: A model with the same structure was trained from scratch for 100 epochs. AT: An 8-neuron adaptation layer was added, trained for 50 epochs, and then the full model was fine-tuned for 50 epochs. The results, averaged over three runs, are summarized in Table 5. Despite the chaotic nature of the system and the severe data limitation, our HTL method achieves the lowest test MSE (0.0426). It performs 1.4 times better than the next-best baseline, FT (0.0599), and 2.7 times better than training from scratch (FC, 0.1153). This result strongly demonstrates that our HTL framework is not limited to one class of problems and can effectively transfer knowledge to novel, complex domains even under conditions of high heterogeneity and data scarcity.

## B.5 ABLATION STUDY

Our proposed HTL method consists of three major stages. First, the pre-trained model is developed on the source dataset. Second, the feature space adaptation is conducted by freezing the pre-trained model and training only the transformation layers with our designed loss function. Finally, the entire model is fine-tuned using the target dataset under MSE loss function.To illustrate the necessity of each major component, four ablation experiments are designed. Case AA (Without Pre-training):

Table 5: Model Performance Comparison for Weather prediction (Mean $\pm$ Std over 3 runs)

| Method | Train | Test |
|--------|-------|------|
| **HTL** | $\mathbf{0.0330 \pm 0.0057}$ | $\mathbf{0.0426 \pm 0.0061}$ |
| FC | $0.0932 \pm 0.0248$ | $0.1153 \pm 0.0209$ |
| FT | $0.0505 \pm 0.0129$ | $0.0599 \pm 0.0136$ |
| AT | $0.1013 \pm 0.0186$ | $0.1168 \pm 0.0188$ |
| LEADS | $0.1807 \pm 0.0475$ | $0.1885 \pm 0.0499$ |

The pre-trained model is removed, and the base model is randomly initialized before proceeding to the adaptation and fine-tuning stages. Case AB (Without Feature Adaptation): The feature adaptation stage is removed. We proceed directly from pre-trained model to fine-tuning the entire model with the MSE loss function. Case AC (Without Fine-tuning): The final fine-tuning stage is removed; training stops after the feature adaptation stage. Case AD (Without Fine-tuning, Extended Adaptation): This case also removes the final fine-tuning step, but extends the epochs for the adaptation to ensure that the overall epochs are the same as the HTL method.

Except for the specific components being ablated, all other settings (e.g., neural network architecture, learning rates, and training epochs for the remaining stages) are identical to those used for the full HTL method to ensure a fair comparison. The results for the ablation studies and the HTL method are summarized in the following table.

The ablation study clearly demonstrates the effectiveness of each major component in our proposed framework. In all cases, removing any stage leads to substantial performance degradation, confirming that every component is critical. Specifically, removing the pre-trained model increases the testing MSE by at least 17%, and by nearly 50% in the water-tank case, underscoring the necessity of leveraging source-domain knowledge.

More critically, removing the feature adaptation stage causes severe performance degradation and even model collapse, highlighting that this stage is the core of our method. For instance, in the CSTR case, the ablation (Case AB) yields the worst performance. Notably, even when the pre-trained model is removed, the source dataset is still used in the designed loss during adaptation, which explains why removing the pre-trained model leads to less degradation than removing the adaptation stage. These results confirm that our designed loss and freeze strategy are essential for effectively adapting pre-trained features to the target domain.

Finally, the study validates the importance of the fine-tuning stage. Omitting it increases the testing error by roughly an order of magnitude for the water-tank and robotics cases, indicating that the model remains under-optimized. Even when we extend adaptation epochs without fine-tuning, the testing MSE remains at least 40% higher than that of the full model.

Table 6: Ablation cases for CSTR, Water tank, and Robotic Cases (Mean $\pm$ Std. All values scaled by $10^{-3}$)

| | | HTL | AA | AB | AC | AD |
|--|--|-----|-----|-----|-----|-----|
| **CSTR** | Train | $\mathbf{2.23 \pm 0.12}$ | $3.10 \pm 0.35$ | $7.57 \pm 0.68$ | $5.13 \pm 0.12$ | $3.50 \pm 0.00$ |
| | Test | $\mathbf{2.47 \pm 0.15}$ | $3.27 \pm 0.40$ | $8.20 \pm 0.96$ | $5.37 \pm 0.15$ | $3.90 \pm 0.00$ |
| **Water Tanks** | Train | $\mathbf{1.00 \pm 0.00}$ | $1.63 \pm 0.40$ | $1.83 \pm 0.49$ | $11.00 \pm 0.00$ | $1.93 \pm 0.06$ |
| | Test | $\mathbf{1.33 \pm 0.06}$ | $1.97 \pm 0.38$ | $2.30 \pm 0.69$ | $12.00 \pm 0.00$ | $2.30 \pm 0.00$ |
| **Robotics** | Train | $\mathbf{5.00 \pm 0.26}$ | $5.20 \pm 0.26$ | $6.93 \pm 0.93$ | $51.00 \pm 9.54$ | $17.67 \pm 1.15$ |
| | Test | $\mathbf{6.13 \pm 0.32}$ | $7.20 \pm 0.00$ | $8.33 \pm 1.12$ | $52.33 \pm 9.02$ | $18.33 \pm 1.53$ |
| **Weather** | Train | $\mathbf{33.0 \pm 5.7}$ | $54.0 \pm 7.3$ | $60.1 \pm 13.5$ | $109.3 \pm 48.2$ | $52.7 \pm 13.2$ |
| | Test | $\mathbf{42.6 \pm 6.1}$ | $77.5 \pm 9.2$ | $69.2 \pm 13.7$ | $112.3 \pm 45.8$ | $60.0 \pm 11.4$ |

# C ROBUSTNESS

## C.1 LIMITED DATA

To validate the performance of our proposed HTL method in the case with limited data size. We selected CSTR and Water tanks cases, with the top two baselines under different number of samples. Specifically, for CSTR, two baselines are selected: FC and AT, and the number of training samples are increased from 50 to 400. For water tank cases FC and FT are selected with samples from 10 to 100. The simulation results are summarized in the following table. As the number of samples decreases, the performance for all the models are degraded. However, the HTL methods still outperform the baselines in both cases. Demonstrating the effectiveness of the proposed HTL strategy. Specifically, for the water tank cases, even with 10 samples, the HTL model still better than that of FC and FT.

Table 7: Mean $\pm$ std of HTL, FC, and AT over three runs for CSTR process(values $\times 10^{-2}$)

| Samples | HTL | FC | AT |
|---|---|---|---|
| 50 | $\mathbf{1.42 \pm 0.10}$ | $2.64 \pm 1.11$ | $2.73 \pm 1.17$ |
| 100 | $\mathbf{1.23 \pm 0.09}$ | $2.64 \pm 0.83$ | $1.92 \pm 0.67$ |
| 200 | $\mathbf{0.67 \pm 0.11}$ | $1.74 \pm 0.42$ | $1.88 \pm 0.66$ |
| 400 | $\mathbf{0.32 \pm 0.01}$ | $0.64 \pm 0.23$ | $0.60 \pm 0.13$ |

Table 8: Mean $\pm$ std of HTL, FC, and FT over three runs for Water tank(values $\times 10^{-2}$)

| Samples | HTL | FC | FT |
|---|---|---|---|
| 10 | $\mathbf{1.76 \pm 0.66}$ | $8.55 \pm 4.00$ | $4.72 \pm 1.59$ |
| 20 | $\mathbf{0.46 \pm 0.20}$ | $5.52 \pm 1.81$ | $3.49 \pm 0.87$ |
| 50 | $\mathbf{0.20 \pm 0.03}$ | $2.05 \pm 1.36$ | $0.59 \pm 0.22$ |
| 100 | $\mathbf{0.15 \pm 0.01}$ | $1.23 \pm 0.40$ | $0.28 \pm 0.06$ |

## C.2 NOISE

We conducted new experiments on the CSTR case study, comparing our HTL method against the top two baselines, FC and FT.We injected Gaussian noise into the normalized training data, while the test set remained noise-free. The noise was modeled with a mean of 0 and a relative standard deviation of 10%, 20%, 40%, and 50%. The (MSE $\pm$ std) results on the test set, averaged over three runs, are in the following table. The results demonstrate the superior robustness of our proposed HTL method. At 10% and 20% noise levels, HTL significantly outperforms both baselines. At the 40% noise level, HTL ($22.40 \times 10^{-3}$) continues to hold a clear advantage over both FC ($36.97 \times 10^{-3}$) and FT ($23.60 \times 10^{-3}$). At the 50% noise extreme, while the performance of all models degrades significantly, our HTL method ($32.30 \times 10^{-3}$) remains highly competitive and nearly identical to the best-performing baseline, FT ($31.60 \times 10^{-3}$). This demonstrates that our two-phase strategy, which leverages a pre-trained model with feature transformation, is inherently more resilient to data noise than baseline methods.

Table 9: Performance vs. Noise Level (Mean $\pm$ Std. All values scaled by $10^{-3}$)

| Noise Level | HTL | FC | FT |
|---|---|---|---|
| 10 | $\mathbf{3.40 \pm 0.50}$ | $8.70 \pm 0.62$ | $6.10 \pm 0.85$ |
| 20 | $\mathbf{7.17 \pm 0.49}$ | $11.77 \pm 1.60$ | $8.80 \pm 0.82$ |
| 40 | $\mathbf{22.40 \pm 1.39}$ | $36.97 \pm 1.50$ | $23.60 \pm 1.40$ |
| 50 | $32.30 \pm 2.29$ | $52.33 \pm 1.23$ | $\mathbf{31.60 \pm 1.22}$ |

## C.3 DIFFERENT FEATURE AUGMENTATION METHOD

To test the robustness of our feature transformation, we conducted a new set of experiments on the CSTR case using five different source-domain feature-mapping strategies, including our original

case (repeating $C_D$), zero-padding $(C_D, \mathbf{0}, T)$, alternative feature repetition $(C_D, T, T)$, and mean-value padding $(C_D, \frac{C_D+T}{2}, T)$. The results in the following table show two key findings: Our proposed HTL method achieved the lowest test MSE in all five mapping scenarios. More importantly, HTL's performance is significantly more stable to the choice of mapping strategy. This robustness is clear in the "Average" row: HTL's average MSE (0.0023) is 3.1 times lower than FT (0.0072) and 2.4 times lower than AT (0.0055). Furthermore, the standard deviation of HTL's performance across these diverse cases (0.0002) is 4.5 times smaller than FT's (0.0009). This new analysis confirms that our method's success is not dependent on a specific feature mapping, as our customized adaptation layers are designed to learn the optimal transformation from various heterogeneous source features.

Table 10: Robustness to Source Feature Mapping (MSE $\mu \pm \sigma$, CSTR case).

| Source Feature Set | HTL | FT | AT |
|---|---|---|---|
| $C_D, C_D, T$ (Original) | $\mathbf{0.0025 \pm 0.0001}$ | $0.0075 \pm 0.0008$ | $0.0052 \pm 0.0007$ |
| $C_D, \mathbf{0}, T$ (Zero-pad) | $\mathbf{0.0020 \pm 0.0001}$ | $0.0059 \pm 0.0002$ | $0.0061 \pm 0.0005$ |
| $C_D, T, \mathbf{0}$ (Zero-pad) | $\mathbf{0.0022 \pm 0.0000}$ | $0.0078 \pm 0.0001$ | $0.0059 \pm 0.0015$ |
| $C_D, T, T$ (Repeat T) | $\mathbf{0.0024 \pm 0.0001}$ | $0.0065 \pm 0.0001$ | $0.0053 \pm 0.0006$ |
| $C_D, \frac{C_D+T}{2}, T$ (Mean-pad) | $\mathbf{0.0022 \pm 0.0001}$ | $0.0085 \pm 0.0001$ | $0.0051 \pm 0.0004$ |
| **Average $\pm$ Std** | $\mathbf{0.0023 \pm 0.0002}$ | $0.0072 \pm 0.0009$ | $0.0055 \pm 0.0004$ |

