# OpenReview forum: "Heterogeneous Transfer Learning with Feature Transformation-Based Adaptation for Modeling Dynamical Systems"
_ICLR.cc/2026/Conference — Submitted to ICLR 2026_

### Official Review · Reviewer_DzLo · 2025-10-26

**Soundness:** 2
**Presentation:** 3
**Contribution:** 2
**Rating:** 4
**Confidence:** 2

**Summary:**

This paper investigates heterogeneous transfer learning for modeling dynamical systems, where the source and target domains have different feature spaces. The authors propose a novel framework that integrates customized adaptation layers into a pre-trained model to enable effective feature transformation across domains. A theoretical analysis is provided to evaluate the generalization performance of the transformed model in the target domain. Building on this foundation, a two-phase training strategy is introduced to further enhance adaptation. Experimental results demonstrate the effectiveness of the proposed method.

**Strengths:**

1. The paper addresses a significant but underexplored problem: the generalization of neural network-based models for complex dynamical systems. Tackling this challenge is valuable and appreciated.
2. The authors provide theoretical guarantees regarding model generalization, which adds rigor to the proposed framework.
3. The designed feature transformation modules for enabling target domain adaptation is simple and efficient.

**Weaknesses:**

1. The theoretical analysis lacks a strong connection to the characteristics of dynamical systems, especially nonlinear dynamics.

2. The experimental validation looks too weak. The paper would benefit from more diverse datasets and comprehensive ablation studies to validate the contribution.

**Questions:**

N/A

---

> ### Author Response · Authors · 2025-11-18
>
> We thank the reviewer for the valuable feedback. In response, we have conducted the suggested ablation study, added new cross-domain experiments to assess generalizability, and clarified the scope and relevance of our theoretical analysis to dynamical systems. We address each point in detail below.
>
> **Ablation study**
>
> To validate the independent contributions of our framework's components, we conducted a comprehensive ablation study across all three datasets.
> Our full method involves three key stages: (1) Pre-training on the source, (2) Feature Adaptation to the target domain, and (3) Fine-tuning on the target dataset. We tested this full model against four ablated versions: AA (No Pre-training): Randomly initialized base model. AB (No Adaptation): Skipped the adaptation stage. We proceed directly from pre-trained model to fine-tuning the entire model with adaptation layers under the MSE loss function. AC (No Fine-tuning): Stopped training after the adaptation stage. AD (No Fine-tuning, Extended Adaptation): Case AC, but with adaptation epochs extended to match the total training epochs of our full HTL method.
>
> **Testing Errors for Ablation Cases**
> (All values are Mean ± Std × 10⁻³)
>
> | Experiment | HTL | AA | AB | AC | AD |
> |---|---|---|---|---|---|
> | **CSTR** | **2.47 ± 0.15** | 3.27 ± 0.40 | 8.20 ± 0.96 | 5.37 ± 0.15 | 3.90 ± 0.00 |
> | **Water Tank** | **1.33 ± 0.06** | 1.97 ± 0.38 | 2.30 ± 0.69 | 12.00 ± 0.00 | 2.30 ± 0.00 |
> | **Robotic** | **6.13 ± 0.32** | 7.20 ± 0.00 | 8.33 ± 1.12 | 52.33 ± 9.02 | 18.33 ± 1.53 |
>
> The ablation study clearly demonstrates the effectiveness of each major component in our proposed framework. In all cases, removing any stage leads to substantial performance degradation, confirming that every component is critical. Specifically, removing the pre-trained model increases the testing MSE by at least 17\%, and by nearly 50\% in the water-tank case, underscoring the necessity of leveraging source-domain knowledge.
>
> More critically, removing the feature adaptation stage causes severe performance degradation and even model collapse, highlighting that this stage is the core of our method. For instance, in the CSTR case, the ablation (Case AB) yields the worst performance. Notably, even when the pre-trained model is removed, the source dataset is still used in the designed loss during adaptation, which explains why removing the pre-trained model leads to less degradation than removing the adaptation stage. These results confirm that our designed loss and freeze strategy are essential for effectively adapting pre-trained features to the target domain.
>
> Finally, the study validates the importance of the fine-tuning stage. Omitting it increases the testing error by roughly an order of magnitude for the water-tank and robotics cases, indicating that the model remains under-optimized. Even when we extend adaptation epochs without fine-tuning, the testing MSE remains at least 58\% higher than that of the full model.
>
> In summary, the results validate our design: pre-training provides a crucial foundation, adaptation performs the critical feature transformation, and fine-tuning optimizes for the target task. The removal of any stage causes a significant performance collapse. We will add this complete ablation study to the manuscript.
>
>
> **Cross-domain experiment**
>
> To demonstrate our method's cross-domain capabilities, we have added a new, challenging experiment on a real-world, chaotic dynamical system: weather prediction.
>
> The task involves transferring a model from New York (JFK) to Los Angeles (LAX) and is a rigorous test of our framework due to
> high heterogeneity (i.e., The source (JFK) dataset lacks pressure (P) data, resulting in different feature spaces.) and extreme data scarcity (i.e., The target (LAX) has only 219 training samples.). The dataset is obtained via Meteostat (https://dev.meteostat.net/).
>
> As shown in the table below, our HTL method achieves the lowest test MSE (0.0426), outperforming the next-best baseline (FT) by ~1.4x (0.0599) and decisively beating AT (0.1168). This new result strongly validates that our framework is not limited to control systems but is effective for complex, real-world domains even under high heterogeneity and data scarcity. We will add this full case study to the appendix.
>
> **Model Performance Comparison for Weather Prediction**
> (Mean ± Std over 3 runs)
>
> | Metric | HTL | FT | FC | AT |
> |---|---|---|---|---|
> | Train | **0.0330 ± 0.0057** | 0.0505 ± 0.0129 | 0.0932 ± 0.0248 | 0.1013 ± 0.0186 |
> | Test | **0.0426 ± 0.0061** | 0.0599 ± 0.0136 | 0.1153 ± 0.0209 | 0.1168 ± 0.0188 |

---

> > ### Author Response · Authors · 2025-11-18
> >
> > **Connection to dynamical system**
> >
> > The reviewer is correct that we do not propose a new, first-principles theory for analyzing nonlinear dynamics itself. Our paper's theoretical contribution is instead to provide the generalization bounds for heterogeneous domain adaptation when applied to the specific class of models used for dynamical systems.
> >
> > The connection is made explicit through our hypothesis class $\mathcal{H}$ (Line 163). Our analysis is not for arbitrary functions; it is tailored specifically to sequence-to-vector models ($h:(\mathbf{x}_1,\dots,\mathbf{x}_t)\rightarrow \mathbf{y}_t$, e.g., RNNs), which is the class of models that captures time-dependent dynamics. This theory is not merely descriptive; it directly guides our algorithm's design, including our customized loss function and two-phase training strategy. We then demonstrate empirically that this theoretically-grounded approach is highly effective, validating it on four challenging nonlinear dynamical systems.

---

### Official Review · Reviewer_8hdV · 2025-10-27

**Soundness:** 2
**Presentation:** 2
**Contribution:** 2
**Rating:** 4
**Confidence:** 4

**Summary:**

This paper introduces a novel framework for heterogeneous transfer learning (HTL) tailored to modeling nonlinear dynamical systems when the source and target domains have mismatched feature spaces.

**Strengths:**

The authors provide a detailed derivation of generalization error bounds for HTL using statistical learning theory and derive a two-phase training strategy, in which the custom loss function incorporates multiple theoretical terms, balancing empirical performance with generalization. The framework is modular and easily integrable into existing neural architectures. Adaptation layers are lightweight and interpretable, making the method scalable and practical.

**Weaknesses:**

1. Remark 1: when the input and output dimensions differ between the source and target domains, how the fact that different features repeated or different strategies to add the zero vectors affects the efficiency of the proposed algorithm?
2. dx in (1) is the dimension of the state vector, and dxs in (2) is the dimension of the model input features of the source domain, how to ensure that Assumption 1 is meaningful?
3. Any function h(\cdot) in the set of hypothesis functions H (Line 163-164 in Page 4): the input of h(\cdot) is any vector in the subspace spanned by x1 to xt, or one of x1 to xt?
4. As shown in (2), Xs, Xt, Ys, and Yt are subspaces. In (4), the authors assume that the corresponding subspaces are equivalent. Are the matrices P and Q nonsingular?
5. What about when replacing the linear transformation in (4) as some nonlinear transformation?

**Questions:**

See Weaknesses.

---

> ### Author Response · Authors · 2025-11-18
>
> We thank the reviewer for the insightful and detailed questions, which helped us clarify our paper's core contributions. We have addressed all five points, providing: new experimental results for feature augmentation robustness (Remark 1), detailed clarifications on our theoretical framework and notation (Assumption 1, $d_x$ vs. $d_{xs}$, $P/Q$ matrices, $h(\cdot)$ input), and a clear justification for our linear transformation design choice. We address each point in detail below.
>
> **Different feature augmentation methods**
>
> We thank the reviewer for this excellent question. To test our method's robustness to different feature augmentation strategies, we conducted a new experiment on the CSTR case. We compared our HTL method against the FT and AT baselines using five different feature mapping strategies (e.g., repeating features, zero-padding, mean-padding). The results are summarized below.
>     The results demonstrate two key advantages of our method:
>     Consistent Superiority: HTL achieves the lowest test MSE (best performance) across all five mapping scenarios.
>     High Robustness: The choice of mapping strategy has a significant impact on the baseline methods, but our HTL method is highly stable. As shown in the "Average ± Std" row, HTL's average MSE ($0.0023$) is ~3.1x lower than FT's, and its performance variance across strategies (std. dev. $0.0002$) is 4.5x smaller. This new analysis confirms that our framework is robust to the choice of augmentation. While baselines are sensitive to this choice, our adaptation layers are effective at learning the optimal transformation regardless, proving the resilience of our approach. We will add this analysis to the Appendix.
>
> **Robustness to Source Feature Mapping (CSTR case)**
> (MSE Mean ± Std)
>
> | Source Feature Set | HTL | FT | AT |
> |---|---|---|---|
> | C_D, C_D, T (Original) | **0.0025 ± 0.0001** | 0.0075 ± 0.0008 | 0.0052 ± 0.0007 |
> | C_D, 0, T (Zero-pad) | **0.0020 ± 0.0001** | 0.0059 ± 0.0002 | 0.0061 ± 0.0005 |
> | C_D, T, 0 (Zero-pad) | **0.0022 ± 0.0000** | 0.0078 ± 0.0001 | 0.0059 ± 0.0015 |
> | C_D, T, T (Repeat T) | **0.0024 ± 0.0001** | 0.0065 ± 0.0001 | 0.0053 ± 0.0006 |
> | C_D, (C_D + T)/2, T (Mean-pad) | **0.0022 ± 0.0001** | 0.0085 ± 0.0001 | 0.0051 ± 0.0004 |
> | **Average ± Std** | **0.0023 ± 0.0002** | 0.0072 ± 0.0009 | 0.0055 ± 0.0004 |
>
> **Regarding $d_x$ (state) vs. $d_{xs}$ (feature):**
>
> We thank the reviewer for spotting this inconsistency, which was caused by a typo in Eq. (1) on our part. The reviewer is correct to distinguish these dimensions. The confusion stems from our typo: the state vector is $\mathbf{x} \in \mathcal{R}^{d_{xv}}$ (the state dimension), not $\mathcal{R}^{d_{x}}$. The state dimension ($d_{xv}$) is not required to be the same as the model's input feature dimension ($d_{xs}$). For example, in the CSTR case, the state vector $\mathbf{x}$ is $\{C_A, C_B, T\}$ (so $d_{xv} = 3$), while the model input feature vector is $\{C_A, C_B, T, Q\}$ (so $d_{xs} = 4$).
>  Therefore, Assumption 1 (which requires $d_{xs} = d_{xt}$ and $d_{ys} = d_{yt}$) is meaningful because it applies only to the feature dimensions of the models, not the state dimensions of the underlying systems. We have corrected the typo in the revised manuscript.
>
> **Input of $h(\cdot)$**
>
>  We thank the reviewer for this clarifying question. The input to $h(\cdot)$ is the entire sequence of $t$ vectors, $(\mathbf{x}_1,\dots,\mathbf{x}_t)$, which belongs to the space $R^{(d_x \times t)}$.
> This sequence-based input is essential for our task of modeling dynamical systems, as the prediction $\mathbf{y}_t$ depends on the history of the previous $t$ steps. The function $h$ therefore acts as a sequence-to-vector model (e.g., as implemented by an RNN). We will revise the text around Line 163 to make this definition explicit and remove any ambiguity.

---

> > ### Author Response · Authors · 2025-11-18
> >
> > **Regarding $P$ and $Q$ (nonsingularity)**
> >
> > We thank the reviewer for this sharp question, which helps clarify the exact role of $P$ and $Q$. The reviewer is correct that $P$ and $Q$ are square (due to Assumption 1). However, Eq. (4) does not assume subspace equivalence. Rather, it defines $P$ and $Q$ as the learnable transformation matrices that map between the spaces. Therefore, $P$ and $Q$ are not required to be nonsingular (invertible). Their function is to serve as the learnable components of our transformed model, $h^*(X_T) = Qh(PX_T)$, as defined in Eq. (5). Our theoretical analysis, including the derivation of the generalization bound in Theorem 1, does not depend on the invertibility of these matrices, only that they are learnable transformations of the appropriate dimensions.
> >
> >  **Nonlinear transformation**
> >
> >  We thank the reviewer for this insightful question. This was a deliberate design choice motivated by two critical factors, one theoretical and one practical.
> >
> > Theoretical Tractability: The linear transformation is essential for our theoretical contribution, as it permits the tractable derivation of the generalization error bound presented in Theorem 1. A nonlinear transformation would make this analysis significantly more complex and is beyond the scope of this paper's core theoretical guarantees.
> >
> > Practical Robustness: Our framework is designed for heterogeneous transfer, where the target domain often has limited data. A complex nonlinear transformation would introduce a large number of new parameters, creating a very high risk of overfitting. The linear approach is a more robust choice, making it far more suitable for these practical, data-scarce scenarios.

---

### Official Review · Reviewer_D1Da · 2025-10-29

**Soundness:** 2
**Presentation:** 3
**Contribution:** 3
**Rating:** 4
**Confidence:** 3

**Summary:**

A method for heterogeneous transfer learning on dynamical systems is proposed. It is based on linear transformations of the inputs and outputs. A bound of the generalization error in the target domain is provided. The proposed loss function is based on the insight obtained from the bound. The experimental results support the utility of the proposed method compared to several baseline methods.

**Strengths:**

- The method is simple.
- The analysis of the generalization error bound may not be overly technically novel but does make sense, and the insight obtained from the bound is neatly utilized in the loss function.

**Weaknesses:**

Although the technical contribution looks solid, the paper seems to need some updates to clear the bar to appear in ICLR.

**(1)**
Most notably, no ablation studies are presented, due to which we cannot analyze how each part of the proposed method was effective. The loss function has multiple terms, and the proposed algorithm comprises two different phases. The contributions of each of these components of the method should be examined in more detail.

**(2)**
The writing in Section 6 could be polished much more. Currently it rephrases the same thing over and over, and it's hard to extract important information.

**(3)**
The necessity of Assumption 1 is unclear. The matrices $P$ and $Q$ can simply be of sizes $d_{xs} \times d_{xt}$ and $d_{xt} \times d_{xs}$, respectively.

Below are minor things:
- Around Eq. (7) (and on the other occasions too), quantities $B_{V,F}$, $B_{W,F}$, $B_{U,F}$ are used without definition.
- Line 345: "The second term measures the performance of $h^*$ on the empirical source and target domains." ... I don't think so, it instead measure the *difference* of the performances.

**Questions:**

Do you have any results of ablation studies, investigating the effect of each component of the method?

---

> ### Author Response · Authors · 2025-11-18
>
> We thank the reviewer for the constructive feedback. In response, we conducted the requested ablation study confirming the necessity of each component, demonstrated that Assumption 1 is a non-restrictive theoretical device, and substantially revised Section 6 while addressing all minor issues. Detailed responses follow below.
>
> **Ablation study**
>
> To validate the independent contributions of our framework's components, we conducted a comprehensive ablation study across all three datasets.
> Our full method involves three key stages: (1) Pre-training on the source, (2) Feature Adaptation to the target domain, and (3) Fine-tuning on the target dataset. We tested this full model against four ablated versions: AA (No Pre-training): Randomly initialized base model. AB (No Adaptation): Skipped the adaptation stage. We proceed directly from pre-trained model to fine-tuning the entire model with adaptation layers under the MSE loss function. AC (No Fine-tuning): Stopped training after the adaptation stage. AD (No Fine-tuning, Extended Adaptation): Case AC, but with adaptation epochs extended to match the total training epochs of our full HTL method.
>
> **Testing Errors for Ablation Cases**
> (All values are Mean ± Std × 10⁻³)
>
> | Experiment | HTL | AA | AB | AC | AD |
> |---|---|---|---|---|---|
> | **CSTR** | **2.47 ± 0.15** | 3.27 ± 0.40 | 8.20 ± 0.96 | 5.37 ± 0.15 | 3.90 ± 0.00 |
> | **Water Tank** | **1.33 ± 0.06** | 1.97 ± 0.38 | 2.30 ± 0.69 | 12.00 ± 0.00 | 2.30 ± 0.00 |
> | **Robotic** | **6.13 ± 0.32** | 7.20 ± 0.00 | 8.33 ± 1.12 | 52.33 ± 9.02 | 18.33 ± 1.53 |
>
> The ablation study clearly demonstrates the effectiveness of each major component in our proposed framework. In all cases, removing any stage leads to substantial performance degradation, confirming that every component is critical. Specifically, removing the pre-trained model increases the testing MSE by at least 17\%, and by nearly 50\% in the water-tank case, underscoring the necessity of leveraging source-domain knowledge.
>
> More critically, removing the feature adaptation stage causes severe performance degradation and even model collapse, highlighting that this stage is the core of our method. For instance, in the CSTR case, the ablation (Case AB) yields the worst performance. Notably, even when the pre-trained model is removed, the source dataset is still used in the designed loss during adaptation, which explains why removing the pre-trained model leads to less degradation than removing the adaptation stage. These results confirm that our designed loss and freeze strategy are essential for effectively adapting pre-trained features to the target domain.
>
> Finally, the study validates the importance of the fine-tuning stage. Omitting it increases the testing error by roughly an order of magnitude for the water-tank and robotics cases, indicating that the model remains under-optimized. Even when we extend adaptation epochs without fine-tuning, the testing MSE remains at least 58\% higher than that of the full model.
>
> In summary, the results validate our design: pre-training provides a crucial foundation, adaptation performs the critical feature transformation, and fine-tuning optimizes for the target task. The removal of any stage causes a significant performance collapse. We will add this complete ablation study to the manuscript.
>
> **Section 6**
>
> We thank the reviewer for this valuable feedback on the writing. We agree that Section 6 was repetitive and have thoroughly rewritten it for clarity and conciseness.
> The revised section removes the redundant descriptions. It now presents a more direct and logical progression of our methodology, making the key information easier to extract. We have updated Section 6 in the new version.
>
> **Necessity of Assumption 1**
>
> The necessity of Assumption 1 is strictly for our theoretical analysis, specifically to derive the generalization error bound in Theorem 1. This bound requires calculating the performance discrepancy between the original model ($h$) and the transformed model ($h^*$), which is only tractable if both models share the same input/output dimensions. We acknowledge that this is a theoretical constraint. As the reviewer notes, in practice, the source and target domains may not share the same number of features. However, as we discuss in Remark 1, this assumption can be easily satisfied in practice using standard feature augmentation methods (e.g., zero-padding) to equalize the feature dimensions before applying the transformations.
>
>
> **Minor Problems**
>
> Regarding the undefined quantities: $B_{V,F}$, $B_{W,F}$, and $B_{U,F}$ denote the upper bounds of the weight parameters in the neural network model $h$, specifically for the output layer ($B_{V,F}$) and the hidden layer (input $B_{W,F}$ and hidden state $B_{U,F}$), respectively. Regarding Line 345: The second term measures the performance difference of $h^*$ on the empirical source and target domains. We will fix these problems in the revised manuscript.

---

### Official Review · Reviewer_FXTu · 2025-10-30

**Soundness:** 3
**Presentation:** 3
**Contribution:** 3
**Rating:** 6
**Confidence:** 3

**Summary:**

This paper proposes a Heterogeneous Transfer Learning framework that aligns source and target feature spaces through linear adaptation layers added before and after a pre-trained model. Experiments showed the effectiveness of the proposed method.

**Strengths:**

1. A novel heterogeneous transfer learning framework is proposed to address the feature mismatch between the source and target domains.
2. Theoretical analysis is provided.
3. Experiments showed the effectiveness of the proposed method.

**Weaknesses:**

1. Insufficient analysis of data scale and noise sensitivity;
2. Limit in the ablation study.

**Questions:**

1. All current experiments have been conducted under conditions with sufficient data volume and controlled noise levels. It remains unclear how the proposed HTL framework performs in scenarios with small sample sizes or high noise levels.
2. The paper proposes a two-stage training process (Phase 1 feature adaptation and Phase 2 global fine-tuning). Still, the current experimental section fails to fully validate the independent contributions and synergistic effects of the two stages.
3. The adaptation matrices P and Q are theoretically responsible for implementing linear mappings between the feature spaces of the source domain and target domain. However,  is there a significant difference among all competing ones?
4. All datasets are from continuous control systems. How is the performance of the proposed method in more cross-domain experiments?

---

> ### Author Response · Authors · 2025-11-18
>
> We thank the reviewer for the valuable feedback. In response, we have added: (i) a robustness analysis for limited-data and high-noise scenarios, (ii) a comprehensive ablation study validating each component of our framework, and (iii) a new cross-domain experiment demonstrating generalizability. We address each specific point in detail below.
>
> **Limited data and noise**
>
> We thank the reviewer for this constructive suggestion. To thoroughly evaluate our framework's robustness, we conducted new experiments to address both points: performance with limited data and under high-noise conditions.
>
> 1. Performance with Limited Data (Data-Scarce Scenario)
>
> We analyzed performance on the CSTR (50-400 samples) and Water Tank (10-100 samples) cases against the top two baselines for each. The results are summarized below. As the tables show, while performance for all methods degrades with fewer samples, our HTL framework consistently and significantly outperforms the baselines across both cases. Notably, in the Water Tank experiment, HTL remains highly effective with only 10 samples (MSE: $1.76 \times 10^{-2}$), achieving 2.7x and 4.8x lower error than the FT and FC baselines, respectively. This demonstrates the framework's effectiveness in data-scarce regimes.
>
>
>
> **CSTR Process Results** (All values are Mean ± std × 10⁻²)
> | Samples | HTL | FC | AT |
> |---|---|---|---|
> | 50 | **1.42 ± 0.10** | 2.64 ± 1.11 | 2.73 ± 1.17 |
> | 100 | **1.23 ± 0.09** | 2.64 ± 0.83 | 1.92 ± 0.67 |
> | 200 | **0.67 ± 0.11** | 1.74 ± 0.42 | 1.88 ± 0.66 |
> | 400 | **0.32 ± 0.01** | 0.64 ± 0.23 | 0.60 ± 0.13 |
>
> **Water Tank Results** (All values are Mean ± std × 10⁻²)
> | Samples | HTL | FC | FT |
> |---|---|---|---|
> | 10 | **1.76 ± 0.66** | 8.55 ± 4.00 | 4.72 ± 1.59 |
> | 20 | **0.46 ± 0.20** | 5.52 ± 1.81 | 3.49 ± 0.87 |
> | 50 | **0.20 ± 0.03** | 2.05 ± 1.36 | 0.59 ± 0.22 |
> | 100 | **0.15 ± 0.01** | 1.23 ± 0.40 | 0.28 ± 0.06 |
>
> 2. Performance with High Noise Levels
>
> We evaluated noise robustness on the CSTR case, injecting Gaussian noise (10\% to 50\% relative standard deviation) into the training data. The results demonstrate our method's superior resilience to noise. HTL significantly outperforms both baselines at 10\%, 20\%, and 40\% noise levels. Even at an extreme 50\% noise level, our method ($32.30 \times 10^{-3}$) remains highly competitive with the best-performing baseline, FT ($31.60 \times 10^{-3}$), and is substantially better than FC ($52.33 \times 10^{-3}$). This confirms that our two-phase strategy, which leverages a pre-trained model with feature transformation, is inherently robust to both data scarcity and significant data noise. We will add these new experiments and a detailed discussion to the Appendix. We thank the reviewer again for prompting these valuable additions, which we believe strengthen the paper.
>
> **Performance vs. Noise Level for CSTR Process**
> (All values are Mean ± Std × 10⁻³)
>
> | Noise Level | HTL | FC | FT |
> |---|---|---|---|
> | 10 | **3.40 ± 0.50** | 8.70 ± 0.62 | 6.10 ± 0.85 |
> | 20 | **7.17 ± 0.49** | 11.77 ± 1.60 | 8.80 ± 0.82 |
> | 40 | **22.40 ± 1.39** | 36.97 ± 1.50 | 23.60 ± 1.40 |
> | 50 | 32.30 ± 2.29 | 52.33 ± 1.23 | **31.60 ± 1.22** |
>
> **Ablation study**
>
> To validate the independent contributions of our framework's components, we conducted a comprehensive ablation study across all three datasets.
> Our full method involves three key stages: (1) Pre-training on the source, (2) Feature Adaptation to the target domain, and (3) Fine-tuning on the target dataset. We tested this full model against four ablated versions: AA (No Pre-training): Randomly initialized base model. AB (No Adaptation): Skipped the adaptation stage. We proceed directly from pre-trained model to fine-tuning the entire model with adaptation layers under the MSE loss function. AC (No Fine-tuning): Stopped training after the adaptation stage. AD (No Fine-tuning, Extended Adaptation): Case AC, but with adaptation epochs extended to match the total training epochs of our full HTL method.
>
> **Testing Errors for Ablation Cases**
> (All values are Mean ± Std × 10⁻³)
>
> | Experiment | HTL | AA | AB | AC | AD |
> |---|---|---|---|---|---|
> | **CSTR** | **2.47 ± 0.15** | 3.27 ± 0.40 | 8.20 ± 0.96 | 5.37 ± 0.15 | 3.90 ± 0.00 |
> | **Water Tank** | **1.33 ± 0.06** | 1.97 ± 0.38 | 2.30 ± 0.69 | 12.00 ± 0.00 | 2.30 ± 0.00 |
> | **Robotic** | **6.13 ± 0.32** | 7.20 ± 0.00 | 8.33 ± 1.12 | 52.33 ± 9.02 | 18.33 ± 1.53 |
>
> The ablation study clearly demonstrates the effectiveness of each major component in our proposed framework. In all cases, removing any stage leads to substantial performance degradation, confirming that every component is critical. Specifically, removing the pre-trained model increases the testing MSE by at least 17\%, and by nearly 50\% in the water-tank case, underscoring the necessity of leveraging source-domain knowledge.

---

> > ### Author Response · Authors · 2025-11-18
> >
> > More critically, removing the feature adaptation stage causes severe performance degradation and even model collapse, highlighting that this stage is the core of our method. For instance, in the CSTR case, the ablation (Case AB) yields the worst performance. Notably, even when the pre-trained model is removed, the source dataset is still used in the designed loss during adaptation, which explains why removing the pre-trained model leads to less degradation than removing the adaptation stage. These results confirm that our designed loss and freeze strategy are essential for effectively adapting pre-trained features to the target domain.
> >
> > Finally, the study validates the importance of the fine-tuning stage. Omitting it increases the testing error by roughly an order of magnitude for the water-tank and robotics cases, indicating that the model remains under-optimized. Even when we extend adaptation epochs without fine-tuning, the testing MSE remains at least 58\% higher than that of the full model.
> >
> > In summary, the results validate our design: pre-training provides a crucial foundation, adaptation performs the critical feature transformation, and fine-tuning optimizes for the target task. The removal of any stage causes a significant performance collapse. We will add this complete ablation study to the manuscript.
> >
> > **P and Q**
> >
> > We designed our ablation study specifically to test this. We compared our full HTL method against Case AB, an ablation where the training stage for matrices P and Q was entirely removed.
> > The results confirm the difference is highly significant. Removing this adaptation stage caused a performance collapse. For instance, on the CSTR test set, the error increased by 3.3-fold (from $2.47 \times 10^{-3}$ for our full HTL to $8.20 \times 10^{-3}$ for Case AB). This demonstrates that our proposed feature adaptation, implemented via matrices P and Q, is not just a theoretical construct but the key component enabling effective knowledge transfer in our framework. This mechanism is precisely what distinguishes our method from, and gives it a significant advantage over, competing baselines.
> >
> > **Cross-domain experiment**
> >
> > To demonstrate our method's cross-domain capabilities, we have added a new, challenging experiment on a real-world, chaotic dynamical system: weather prediction.
> >
> > The task involves transferring a model from New York (JFK) to Los Angeles (LAX) and is a rigorous test of our framework due to
> > high heterogeneity (i.e., The source (JFK) dataset lacks pressure (P) data, resulting in different feature spaces.) and extreme data scarcity (i.e., The target (LAX) has only 219 training samples.). The dataset is obtained via Meteostat (https://dev.meteostat.net/).
> >
> > As shown in the table below, our HTL method achieves the lowest test MSE (0.0426), outperforming the next-best baseline (FT) by ~1.4x (0.0599) and decisively beating AT (0.1168). This new result strongly validates that our framework is not limited to control systems but is effective for complex, real-world domains even under high heterogeneity and data scarcity. We will add this full case study to the appendix.
> > **Model Performance Comparison for Weather Prediction**
> > (Mean ± Std over 3 runs)
> >
> > | Metric | HTL | FT | FC | AT |
> > |---|---|---|---|---|
> > | Train | **0.0330 ± 0.0057** | 0.0505 ± 0.0129 | 0.0932 ± 0.0248 | 0.1013 ± 0.0186 |
> > | Test | **0.0426 ± 0.0061** | 0.0599 ± 0.0136 | 0.1153 ± 0.0209 | 0.1168 ± 0.0188 |

---

### Author Response · Authors · 2025-11-28

Dear Area Chair and Reviewers,

We sincerely thank you for your constructive feedback. We have just uploaded a revised version of our manuscript. Key updates include:

**New Ablation Studies**: We have added comprehensive ablation studies for all experimental cases, quantitatively verifying the necessity of each stage in our framework (Pre-training, Adaptation, Fine-tuning).

**New Complex Case Study**: We added a "Weather Prediction" case study to demonstrate the generalizability of our method on a chaotic, real-world system.

**Robustness Analysis (Appendix C)**: We added extensive sensitivity analyses regarding limited data (down to 10 samples), high noise levels (up to 50%), and various feature augmentation strategies.

**Methodology Clarification (Section 6)**: We have rewritten Section 6 to be more concise and precise, as requested. we have also added the discussion regarding the nonlinear transformation methods and the theoretical analysis.

We believe these additions significantly strengthen the paper and address the concerns raised regarding robustness and component validation. We welcome any further questions you may have in the remaining time.

Best regards,

The Authors

---

### Author Response · Authors · 2025-12-02
**Summary of Revisions for the Area Chair**

Dear Area Chair,

We are posting this summary to assist you in evaluating our revised manuscript. Although we could not conclude the discussion with the reviewers, we have conducted the specific new experiments they requested to address their concerns regarding component validity, generalizability, and robustness.

Below is a summary of the major updates:

**1. Validation of Framework Components (Reviewers FXTu, D1Da, DzLo)**

**Concern:** Reviewers asked if the separate training stages (Pre-training, Adaptation, Fine-tuning) were necessary or what was the contributions of the component of the two-stage training process.

**Our Revision:** We added a **full ablation study (Section 7.5)** across all datasets.

**Result:** The experiments show that removing the adaptation stage (**Case AB**) leads to a significant performance drop. This confirms that our proposed adaptation matrices are **essential for effective transfer**, rather than redundant.

**2. Generalizability to Complex Systems (Reviewers FXTu, DzLo)**

**Concern:** Questions were raised about whether the framework performs well beyond standard, controlled dynamical systems.

**Our Revision:** We introduced a **Weather Prediction case study (Section 7.4)** to test the method on a chaotic, real-world system.

**Result:** Our HTL method **significantly outperforms baselines** in this domain, demonstrating that the linear feature transformation remains effective even for complex, chaotic dynamics.

**3. Robustness Checks (Reviewers FXTu, 8hdV)**

**Concern:** Performance was unclear regarding data scarcity, high noise levels, and sensitivity to feature augmentation.

**Our Revision:** We added comprehensive sensitivity analyses in **Appendix C**.

**Result:**

* **Limited Data:** The model outperforms baselines with as few as **10 training samples**.
* **Noise:** The method remains resilient even with **50% relative noise** added.
* **Augmentation:** Performance is stable across 5 different feature mapping strategies.

**4. Theoretical Clarifications (Reviewers D1Da, 8hdV, DzLo)**

**Concern:** Questions regarding the choice of linear transformations and the necessity of Assumption 1.

**Our Revision:** We clarified that the linear transformation is a deliberate design choice to prevent overfitting in data-scarce target domains. We also added experiments (**Remark 1**) verifying that Assumption 1 is a theoretical tool for derivation and does not limit practical implementation.

We believe these substantial updates directly answer the scientific questions identified during the review. We respectfully request that you consider these new results in your final assessment.

Best regards,
The Authors

---

### Meta-Review · Area_Chair_2SK5 · 2025-12-28

**Summary:**

This submission proposes a heterogeneous transfer learning framework for nonlinear dynamical system modeling under feature-space mismatch between source and target. The method wraps a pre-trained sequence model with lightweight, learnable linear adaptation layers (input/output feature transformations) and trains them via a two-phase strategy (feature adaptation followed by global fine-tuning), with a custom objective motivated by a domain-adaptation generalization analysis. The paper’s intended contribution is a simple, modular adaptation mechanism plus a theoretical bound that guides the training design, supported by experiments on dynamical-system case studies (with additional robustness/ablation and a weather-transfer case added in the revision/rebuttal).

**Reviewer Concerns:**

Across reviewers, the main pre-rebuttal concerns were (i) insufficient experimental validation (limited dataset diversity, unclear robustness to limited data/noise, and missing ablations for the multi-term loss and two-stage training), (ii) clarity and correctness issues in the presentation (notably Section 6’s repetitiveness, notation/definition gaps, and confusion around dimension assumptions), and (iii) a perceived weak linkage between the theoretical analysis and dynamical-systems-specific structure (especially nonlinear dynamics), raising questions about the strength and novelty of the theoretical contribution. The rebuttal substantively addresses several empirical and clarity gaps by adding an ablation study, sensitivity analyses for data scarcity and noise, additional discussion on Assumption 1 and the role of linear transformations, and a new cross-domain weather prediction transfer experiment. However, even after these additions, the overall evidential package remains borderline for ICLR: the empirical coverage is still modest relative to the claims (with limited breadth of real-world dynamical domains and no strong demonstration that the approach materially advances beyond straightforward linear adapter-style domain alignment), and the theoretical component remains largely a general domain-adaptation style argument whose dynamical-systems specificity is limited. Given the remaining uncertainty about significance/novelty and the still-limited validation for a top-tier venue, I recommend rejection.

**Reviewer Scores:**

Before rebuttal, the ratings were FXTu: 6 (confidence 3), D1Da: 4 (confidence 3), 8hdV: 4 (confidence 4), and DzLo: 4 (confidence 2). I estimate the rebuttal would plausibly move D1Da and 8hdV to around 5 and DzLo to around 5 due to the added ablations/clarifications and the new cross-domain experiment, while FXTu would likely remain around 6 (at most a modest increase) given that the core concerns shift from “missing evidence” to “overall strength and scope.” Aggregating these estimated updates yields a borderline profile that still does not clearly meet the acceptance bar, supporting a Reject decision.

---

### Decision · Program_Chairs · 2026-01-26

Reject